# An electrophysiological and kinematic model of *Paramecium*, the "swimming neuron"

**Irene Elices**[1], **Anirudh Kulkarni**[1,2], **Nicolas Escoubet**[3], **Léa-Laetitia Pontani**[3], **Alexis Michel Prevost**[3], **Romain Brette**[1]*

1 Sorbonne Université, INSERM, CNRS, Institut de la Vision, Paris, France, 2 Department of Bioengineering and Centre for Neurotechnology, Imperial College London, South Kensington Campus, London, United Kingdom, 3 Sorbonne Université, CNRS, Institut de Biologie Paris-Seine (IBPS), Laboratoire Jean Perrin (LJP), Paris

* romain.brette@inserm.fr

**Data Availability Statement:** Code for electrophysiology experiments, model fitting and figures can be found at https://github.com/romainbrette/Paramecium-model.

## Abstract

*Paramecium* is a large unicellular organism that swims in fresh water using cilia. When stimulated by various means (mechanically, chemically, optically, thermally), it often swims backward then turns and swims forward again in a new direction: this is called the avoiding reaction. This reaction is triggered by a calcium-based action potential. For this reason, several authors have called *Paramecium* the "swimming neuron". Here we present an empirically constrained model of its action potential based on electrophysiology experiments on live immobilized paramecia, together with simultaneous measurement of ciliary beating using particle image velocimetry. Using these measurements and additional behavioral measurements of free swimming, we extend the electrophysiological model by coupling calcium concentration to kinematic parameters, turning it into a swimming model. In this way, we obtain a model of autonomously behaving *Paramecium*. Finally, we demonstrate how the modeled organism interacts with an environment, can follow gradients and display collective behavior. This work provides a modeling basis for investigating the physiological basis of autonomous behavior of *Paramecium* in ecological environments.

## Author summary

Behavior depends on a complex interaction between a variety of physiological processes, the body and the environment. We propose to examine this complex interaction in an organism consisting of a single excitable and motile cell, *Paramecium*. The behavior of *Paramecium* is based on trial and error: when it encounters an undesirable situation, it backs up and changes direction. This avoiding reaction is triggered by an action potential. Here we developed an empirically constrained biophysical model of *Paramecium*'s action potential, which we then coupled to its kinematics. We then demonstrate the potential of this model in investigating various types of autonomous behavior, such as obstacle avoidance, gradient-following and collective behavior.

Electrophysiology data and analyses, including particle image velocimetry can be found at https://doi.org/10.5281/zenodo.6074166. Behavioral data and code can be found at https://doi.org/10.5281/zenodo.6074480.

**Funding:** This work was supported by Agence Nationale de la Recherche (ANR-20-CE30-0025- 01 to RB, LLP, AMP and ANR-21-CE16-0013-02 to RB), Programme Investissements d'Avenir IHU FOReSIGHT (Grant ANR-18-IAHU-01 to RB), Fondation Pour l'Audition (Grant FPA RD-2017-2 to RB), CNRS (Défi Mécanobiologie, project PERCEE, to RB, LLP and AMP), and Sorbonne Université (Emergence, project NEUROSWIM to RB, LLP and AMP). The funders had no role in study design, data collection and analysis, decision to publish, or preparation of the manuscript.

**Competing interests:** The authors have no competing interests.

## Introduction

Behavior depends on a complex interaction between a variety of physiological processes, the body and the environment. This complexity makes it challenging to develop integrative models relating the different components. Thus, a strategy is to study model organisms that are experimentally accessible and structurally simpler than vertebrates. This strategy has been applied in particular to *C. Elegans*, with its 302 neurons and a known connectome [1–3]. However, electrophysiology is difficult owing to the small size of its neurons (about 3 μm), and developing empirically valid neuromechanical models of *C. Elegans* remains challenging. More recently, other model organisms have been introduced: *Hydra*, with a few thousand neurons and the advantage of being transparent [4,5], and jellyfish *Aurelia aurita* [6].

Here we propose to develop an integrative model of *Paramecium tetraurelia* [7], which is structurally much simpler than the abovementioned model organisms (Fig 1A), since it is a unicellular organism, while being large enough to perform intracellular electrophysiology (about 120 μm long [8]). *Paramecium* is a common ciliate, which swims in fresh water by beating its ~4000 cilia (Fig 1B) [8–10], and feeds on smaller microorganisms (bacteria, algae, yeast). It uses chemical signals to find food (Fig 1C and 1D), avoids obstacles thanks to mechanosensitivity (Fig 1E), displays collective behavior, adapts to changing environmental conditions and can even learn to respond to new stimuli [11].

More than a century ago, Jennings described the basis of its behavior as "trial-and-error" [14]. *Paramecium* normally swims in a helicoidal fashion at about 500–1000 μm/s, but when it encounters something undesirable (obstacle, hot region, noxious substance), it produces an *avoiding reaction* (Fig 2A): it briefly swims backward, then turns and swims forward in a new direction. The avoiding reaction is triggered by a calcium-based graded action potential, which can be observed in an immobilized cell in response to a current pulse (Fig 2B). The calcium current is produced by L-type calcium channels located in the cilia [15], related to the $Ca_V1$ family found in neurons, heart and muscles of mammals [16]. Genes for many

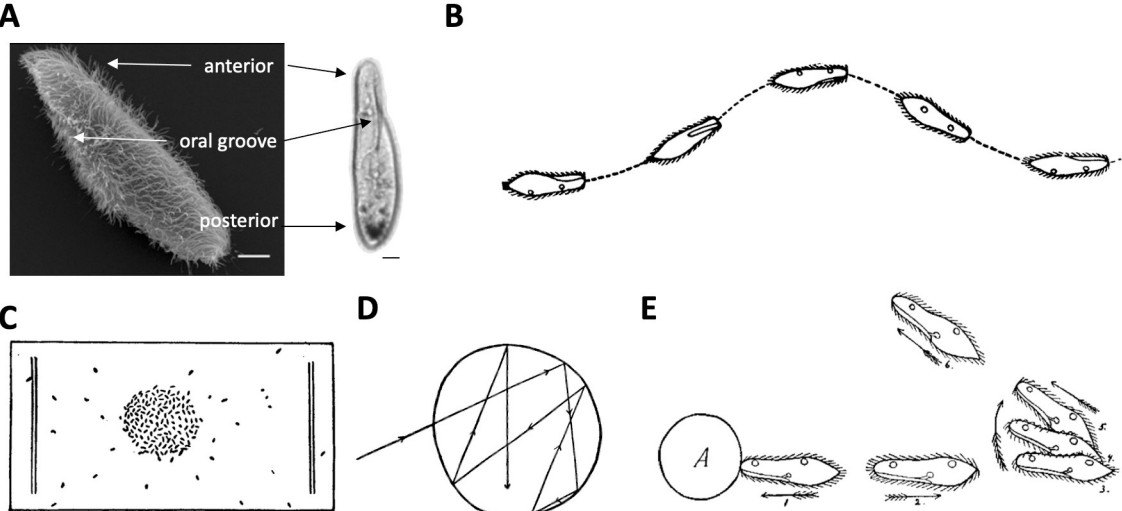

**Fig 1. Presentation of Paramecium.** A, Scanning electron microscopy image (left) [12], and bright field microscopy image (right) of Paramecium tetraurelia (scale bars: 10 μm). B, Typical helicoidal swim of Paramecium [13]. C, Accumulation of paramecia in a drop of acid [13]. D, Trajectory of a single cell in the acid drop, showing directional changes at the boundary [13]. E, Avoiding reaction against an obstacle, showing ciliary reversal followed by reorientation [14].

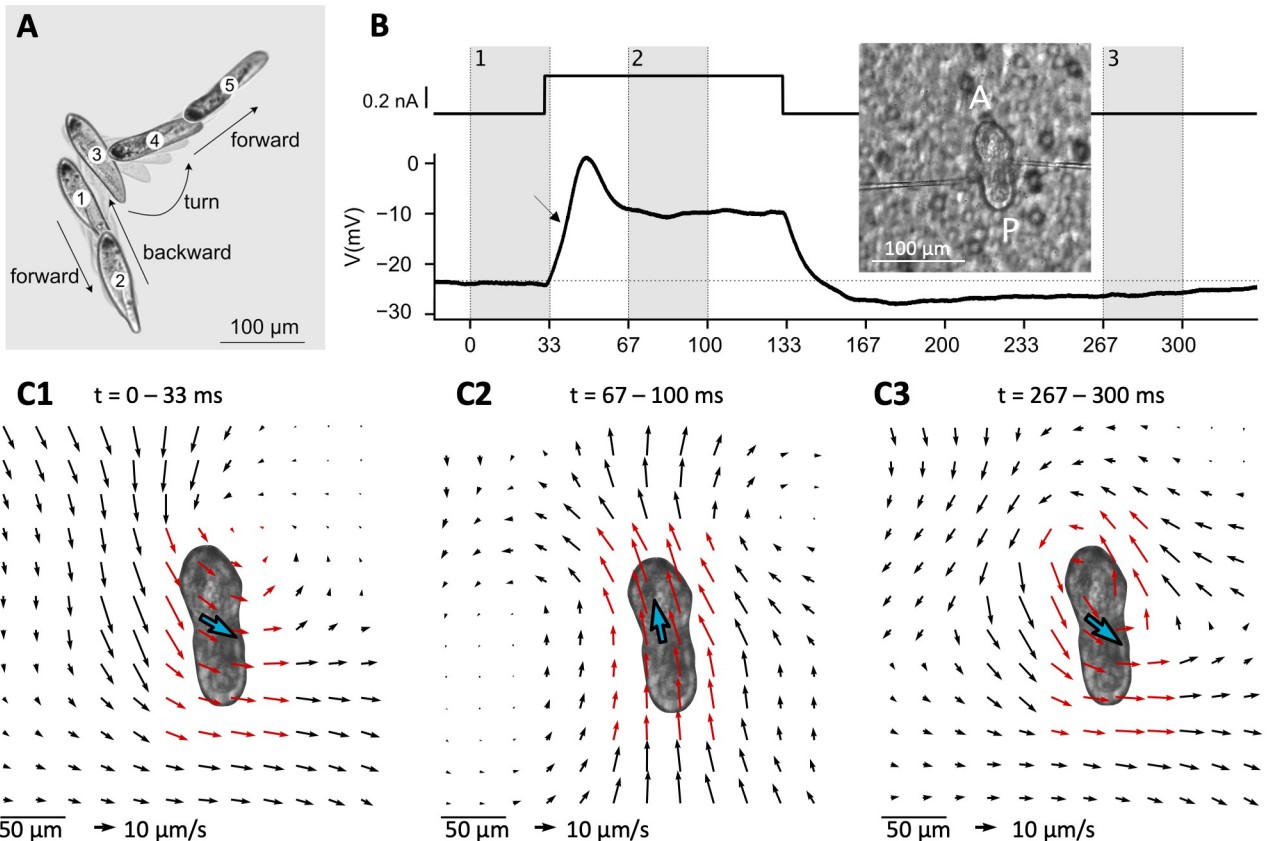

**Fig 2. The avoiding reaction of Paramecium.** A, Typical spontaneous avoiding reaction: the ciliate swims backward, then turns and eventually resumes forward swimming, while spinning around its main axis during the entire movement. Images are separated by 150 ms, with intermediate shaded frames every 37 ms. The cell was placed in 20 mM NaCl and 0.3 mM CaCl₂ to induce spontaneous avoiding reactions [21]. B, Intracellular recording of a voltage response (bottom right) to a square current pulse of amplitude 300 pA (top right) in an immobilized cell (left; A: anterior end; P: posterior end), showing a small action potential (in the standard extracellular solution, see Methods). The arrow points at a small upward inflexion due to the calcium current. Inset: Paramecium immobilized on a filter (background) with two electrodes. C, Velocity field of the fluid on a plane ~30 μm above the cell, calculated over the three shaded intervals shown in B. The blue arrow indicates mean velocity in the whole field, represented twice larger for clarity. The red arrows highlight the area neighboring the cell. C1, The fluid moves backward, which would make the cell swim forward. C2, The fluid moves forward. C3, The flow direction reverts on the posterior end, but not on the anterior right end, resulting in a swirling pattern.

ionic channels have been found in the fully sequenced genome [17], and a number have been electrophysiologically identified [18]. Many signaling pathways of neurons have been found in *Paramecium* [19]. For this reason, *Paramecium* has been called a "swimming neuron" [20] and there is a vast amount of information about its electrophysiology, from studies done mainly in the 1960-80s [15,18]. However, there is no empirically based model of its action potential.

We developed a biophysical model of *Paramecium*'s action potential, based on electrophysiological experiments on immobilized paramecia. We then augmented it by coupling ciliary calcium concentration with kinematic variables, with a phenomenological model constrained by simultaneous imaging of fluid motion induced by cilia beating and measurements of trajectories of freely swimming cells. In this way, we obtain a model of a swimming *Paramecium*, which exhibits spiraling movements and graded avoiding reactions. Finally, we show how the model can be used to investigate closed-loop behavior, including chemotaxis and collective behavior.

## Results

### A brief overview of *Paramecium*'s action potential

In this section, we recapitulate known facts about *Paramecium's* electrophysiology, while illustrating our experimental techniques. To perform intracellular electrophysiology (see Methods, *Electrophysiology*), it is necessary to first immobilize the cell. To this end, we use a device we previously developed [22], which uses a transparent filter with holes smaller than the cell and a peristaltic pump. The pump draws the extracellular solution (4 mM KCl and 1 mM CaCl$_2$) from an outlet below the filter, immobilizing the cells against the filter (Fig 2B, inset). Two high-resistance electrodes are then inserted into the cell and the pump is stopped. The cell is then held by the electrodes. One electrode is used to inject current, while the other is used to measure the membrane potential. *Paramecium* is a large cell (about 120 μm long and 35 μm wide for *P. tetraurelia*), which makes it isopotential [23,24].

Depolarization opens voltage-gated calcium channels located in the ~4000 cilia [8–10], similar to the L-type Cav1.2 family in mammals [16]. This can be noticed on Fig 2B as a small upward deflection before the peak of the membrane potential (arrow). This calcium current, denoted as I$_{Ca}$, activates rapidly (a few milliseconds), producing a current of up to about 4 nA [25]. Calcium entry then makes cilia reorient, which makes them beat forward.

To observe the beating direction, we use 1 μm tracer particles and estimate their velocity with particle image velocimetry (PIV, see Methods), on a plane about 30 μm above the cell (Fig 2C1–2C3, note that fluid velocity is much smaller than swimming velocity as it is measured above the cell [26]). Before the stimulation, the cilia beat backward to the right (Fig 2C1), which tends to make the cell move forward, with a spiraling movement over to the left [27,28], with the oral groove facing the spiral axis [13,29]. When calcium enters the cilia and calcium concentration reaches about 1 μM [30], cilia reorient and beat forward, slightly to the left (Fig 2C2), which makes the cell move backward. The calcium channels inactivate rapidly (a few milliseconds) through calcium-mediated inactivation: intraciliary calcium binds to calmodulin, which then inactivates the channels [18,31,32]. Calcium is then expelled by diffusion, buffering and pumps, in particular plasma membrane calcium pumps (PMCA) identified in the cilia [33,34]. After calcium has entered the cilia, voltage-gated K$^+$ channels located in the basal membrane rapidly open, producing a delayed rectifier current I$_{Kd}$ that damps the membrane potential (Fig 2B, just after the peak). Calcium also activates a smaller K$^+$ current I$_{K(Ca)}$, which can be seen in the prolonged hyperpolarization after the stimulation [35,36].

After the stimulation, when calcium concentration has decreased below ~1 μM [30], cilia revert to backward beating (Fig 2C3). We notice a swirling pattern on Fig 2C3, which can be attributed to an asynchronous reversal of different groups of cilia. We will show how this relates to the change in swimming direction seen on Fig 2A.

We used current-clamp recordings and PIV measurements of fluid motion to build a model of the action potential together with electromotor coupling, summarized in Fig 3. We chose to use current-clamp rather than voltage-clamp recordings because good control is difficult to achieve with high resistance electrodes and several important processes are calcium-gated rather than voltage-gated, making voltage-clamp less relevant. Our modeling strategy is outlined in Fig 4. We start by determining passive properties (Fig 5) by using responses to hyperpolarized pulses, which do not trigger the calcium-based avoiding reaction. Then we isolate the delayed rectifier current I$_{Kd}$ mechanically by deciliating the cells, thereby removing the voltage-gated calcium channels (Fig 6). We use these findings to build a complete model of the action potential with calcium dynamics coupled with ciliary reversal (as sketched in Fig 3), which we fit to electrophysiological responses of ciliated cells together with measurements of fluid motion (Fig 7). To couple this electrophysiological model with kinematics, we examine a

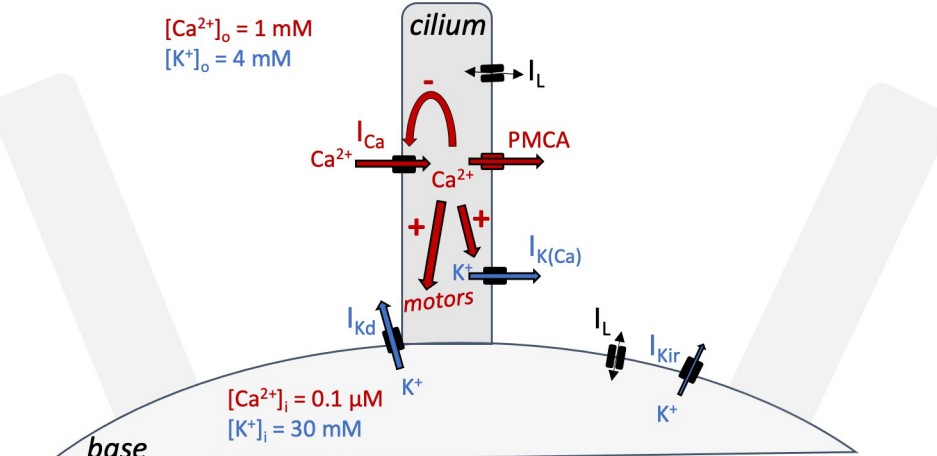

**Fig 3. Summary of the model, showing a cilium attached to the base, and the movements of the two main ions, $Ca^{2+}$ (red) and $K^+$ (blue).** There is more $Ca^{2+}$ outside than inside, and more $K^+$ inside than outside. Calcium enters through ciliary voltage-gated channels as a current $I_{Ca}$. It then quickly inactivates these channels, forming a negative feedback loop. Calcium activates motor proteins, triggering ciliary reversal, as well as a ciliary $K^+$ channel, producing an outward $K^+$ current $I_{K(Ca)}$. The motor activation results in calcium concentration $[Ca^{2+}]_i$ modulating kinematic parameters. Calcium is then expelled, in particular by plasma membrane $Ca^{2+}$ ATPases (PMCA). Depolarization opens voltage-gated $K^+$ channels in the basilar membrane, presumably near the cilium, creating a current $I_{Kd}$ (delayed rectifier). A linear leak current is also present. Finally, an inward rectifier current $I_{Kir}$ opens at very hyperpolarized voltages, which has little impact on the avoiding reaction.

simplified hydrodynamic model where cilia reverse with heterogeneous calcium sensitivity, as suggested by our PIV measurements and previous studies (Fig 8). This study allows us to infer the form of couplings between calcium concentration and kinematic variables, which we partially constrain with behavioral measurements of freely swimming cells (Fig 9). With these couplings, the electrophysiological model can then be used to simulate free swimming and behavioral responses to stimuli (Fig 9). Finally, we augment the model with elementary models of transduction and show simulations of behavior in structured environments, such as obstacle avoidance (Fig 10) and gradient following (Fig 11).

## Passive properties

We start by estimating the passive properties (resistance, capacitance, reversal potentials) with model fitting techniques (see Methods, *Electrophysiological modeling* and *Model optimization*), and we compare with previous measurements in the literature. To this end, we use voltage responses to hyperpolarizing current pulses (duration 100 ms, amplitude 0 to -4 nA in 300 pA increments; Fig 5A, top). Such stimuli are known to trigger different voltage-gated currents: a small inactivating calcium current [37–39], and a strong inward rectifier $K^+$ current $I_{Kir}$, i.e., which lets current pass mainly in the inward direction, below the reversal potential of $K^+$ [40,41]. None of these currents are involved in the avoiding reaction and the action potential, and therefore they will not be included in the final model. In particular, the small calcium current is related to the escape reaction, an increase in swimming speed triggered by hyperpolarizing stimuli [42,43]. However, the inward rectifier current will allow us to estimate $E_K$, the reversal potential of $K^+$.

When the pulse intensity is strong, an inward current can be seen to activate after ~15 ms. The inward rectifier current has the property to activate mainly below $E_K$ [44]. This can be seen by removing $K^+$ from the extracellular solution (making $E_K = -\infty$). With 4 mM

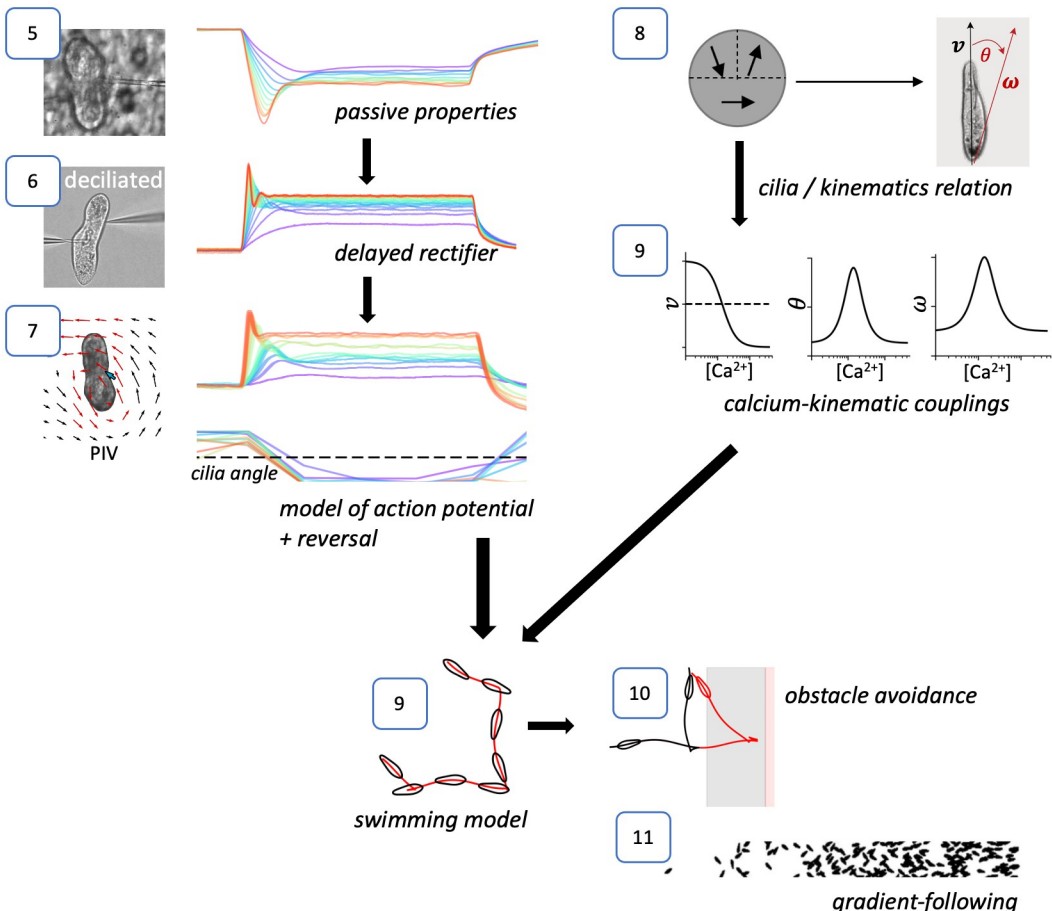

**Fig 4. Outline of the modeling strategy.** Each number indicates the corresponding figure. The left column (5–7) is the electrophysiological modeling. The right column (8–9) is the modeling of couplings between electrophysiology and kinematic variables. Put together, we obtain a model of a freely swimming Paramecium (9), which is then augmented with elementary sensory transduction models to yield model simulations in structured environments (10–11).

extracellular KCl, hyperpolarization below about -60 mV activates a strong inward current, which is largely suppressed in 0 mM KCl (Fig 5B). After the pulse, the K$^+$ current switches from inward to outward as it passes $E_K$. This results in a change in decay speed, which is noticeable at about -60 mV in Fig 5A (top, arrow). We use this property to estimate $E_K$.

To this end, we fit a biophysical model consisting of a linear leak current and an inward rectifier current $I_{Kir}$ (Fig 5A, bottom) (see Methods, *Electrophysiological modeling*) using the Brian 2 simulator [45] with the model fitting toolbox [46], which applies differential evolution and gradient descent for least square estimation of model parameters.

We modeled the inward rectifier current as a non-inactivating current with Boltzmann activation, two gates and a fixed time constant: $I_{Kir} = g_{Kir} n_{Kir}^2 (E_K - V)$ (Eqs (4) and (5); $g_{Kir}$ is maximal conductance and $n_{Kir}$ is the gating variable). Fig 5C shows the activation curve $n_{Kir}^2(V)$ of the cell shown in Fig 5A (dashed) and the curve with median parameters (solid), which confirms that the current activates essentially below $E_K$.

We find that the resting potential $V_0$ is -24.5 mV ± 10.6 mV (n = 28; median -22.5 mV; Fig 5D). Oertel et al. [25] previously reported about -23 mV with a slightly different extracellular solution. Capacitance $C$ is 289 ± 75 pF. By comparison, *P. caudatum*, which is larger, has a

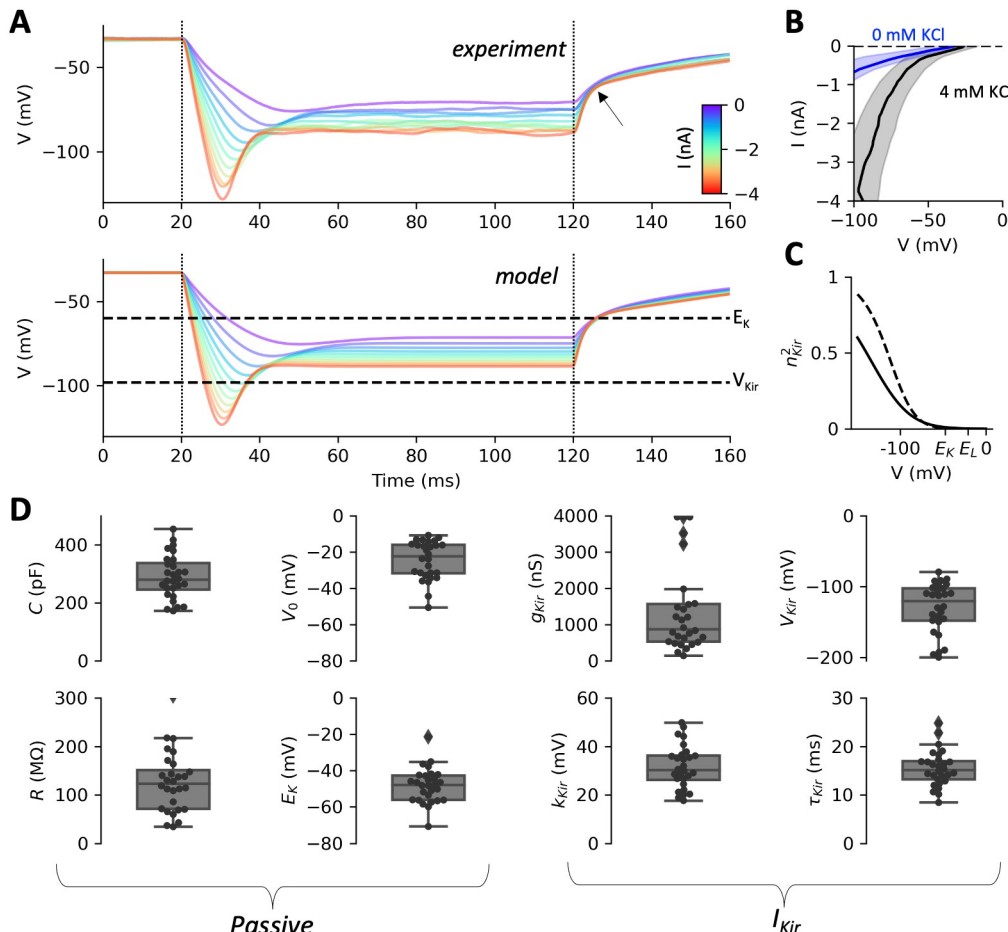

**Fig 5. Passive properties and inward rectifier current.** A, Top: voltage responses of one cell to negative current pulses (I = 0 to -4 nA in 300 pA increments; dashed lines: start and end of pulses), in the standard extracellular solution (4 mM KCl and 1 mM CaCl$_2$). The arrow points at an inflexion due to the inward rectifier current I$_{Kir}$. Bottom: model responses fitted to the data, showing the inferred reversal potential of K$^+$ (E$_K$) and the half-activation voltage V$_{Kir}$ of the inward rectifier current. B, Current-voltage relationship over all cells (mean ± standard deviation, measured at pulse end) in 4 mM KCl (grey) and 0 mM KCl (blue). Removing K$^+$ from the extracellular solution largely suppresses the inward current. C, Activation curve of the inward rectifier current in the fitted models. The current activates below E$_K$ (E$_L$ is leak reversal potential). The solid curve is the activation function with median parameters, the dashed curve is the activation function of the cell shown in A. D, Fitted parameters over n = 28 cells, grouped in passive parameters and inward rectifier parameters.

capacitance of about 700 pF. Since *P. caudatum* is 200 μm long and 46 μm wide [47] and *P. tetraurelia* is 115 μm long and 34 μm wide [8], a simple scaling would predict a capacitance (200 x 46) / (115 x 34) ≈ 2.35 times smaller for *P. tetraurelia*, i.e. about 300 pF, which is consistent with our estimates.

Resistance at rest R (including the contribution from I$_{Kir}$) is 126 ± 62 MΩ. Finally, we find E$_K$ = -48 ± 10 mV, corresponding to an intracellular K$^+$ concentration [K$^+$]$_i$ = 29 ± 11 mM (based on the Nernst equation and given that [K$^+$]$_o$ = 4 mM). This is consistent with estimates in the literature obtained with various methods, varying between 18 and 34 mM [40,48–50].

We briefly describe the parameter estimation results for I$_{Kir}$, even though these will not be further used, as only E$_K$ plays a role in depolarized responses. The estimates for total conductance g$_{Kir}$ and half-activation voltage V$_{Kir}$ are variable across cells, presumably because these

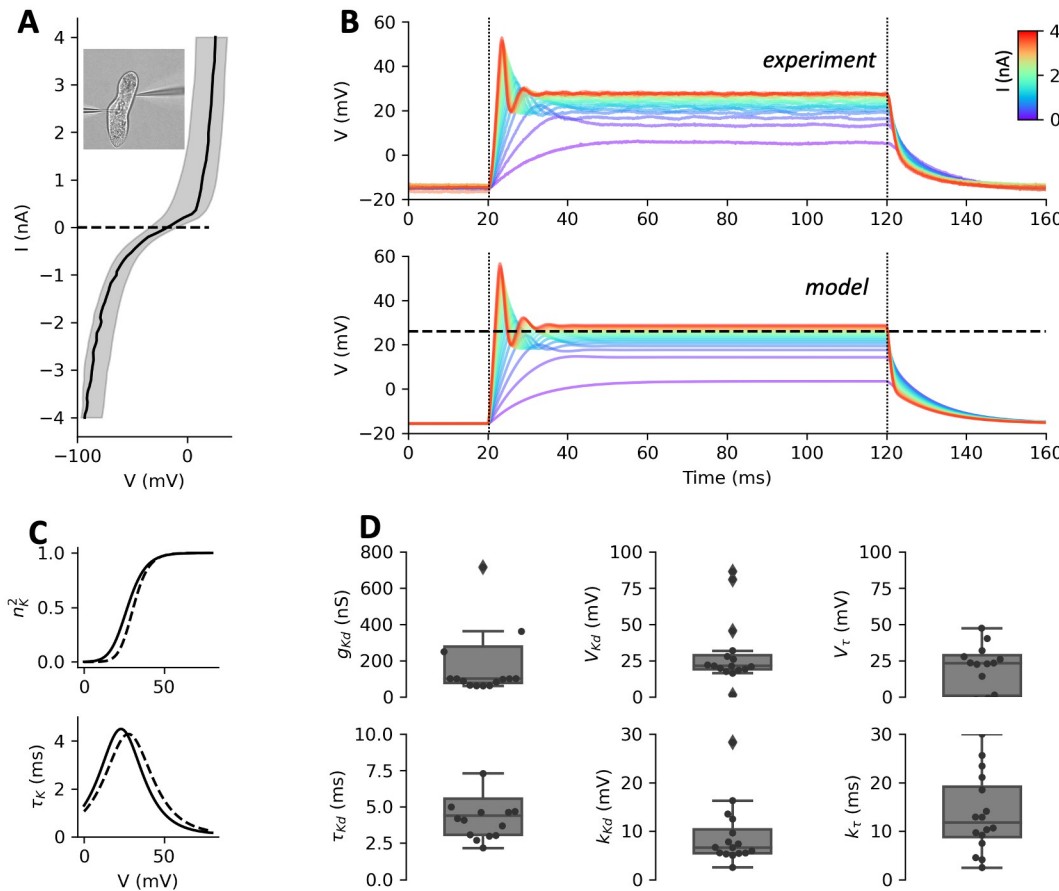

**Fig 6. The delayed rectifier current measured in deciliated cells.** A, Current-voltage relationship in deciliated cells, showing a strong delayed rectifier current for depolarized voltages, and the inward rectifier current for hyperpolarized voltages. B, Top: voltage responses of one cell to positive current pulses (I = 0 to 4 nA in 300 pA increments). Bottom: responses of the two-gate Boltzmann model fitted to the data, showing the inferred half-activation voltage of the delayed rectifier current (dashed). C, Activation and time constant of the delayed rectifier current as a function of voltage in fitted models, with median parameters (solid) and for the cell shown in B (dashed). D, Statistics of fitted parameters (n = 16).

parameters are not well constrained by the data ($g_{Kir}$ and $V_{Kir}$ cannot be estimated independently in the voltage region where channels are mostly closed). Nonetheless, the results confirm that $I_{Kir}$ activates essentially below $E_K$. Activation slope ($k_{Kir} = 32 \pm 9$ mV) and time constant ($\tau_{Kir} = 16 \pm 4$ ms) are better constrained. With the estimated parameters, the inward rectifier current contributes about 14% of the resting conductance (median; $16 \pm 14\%$).

## A model of the deciliated cell

Next, we analyzed the delayed rectifier current $I_{Kd}$ responsible for repolarization. The ciliary calcium channel can be pharmacologically blocked with W-7, but this drug is toxic to *Paramecium* [51]. Instead, we isolated the delayed rectifier current mechanically by removing the cilia with ethanol [47,52] (see Methods, *Deciliation*). This procedure does not kill the cell, and cilia grow back after a few hours. It removes the voltage-gated calcium channels, which are located in the cilia [16,47], and thereby also removes the calcium-activated K⁺ current $I_{K(Ca)}$. In addition, it is no longer necessary to use the immobilization device (Fig 6A, inset). As can be seen on Fig 3, this procedure should leave only the delayed rectifier current $I_{Kd}$, activated upon

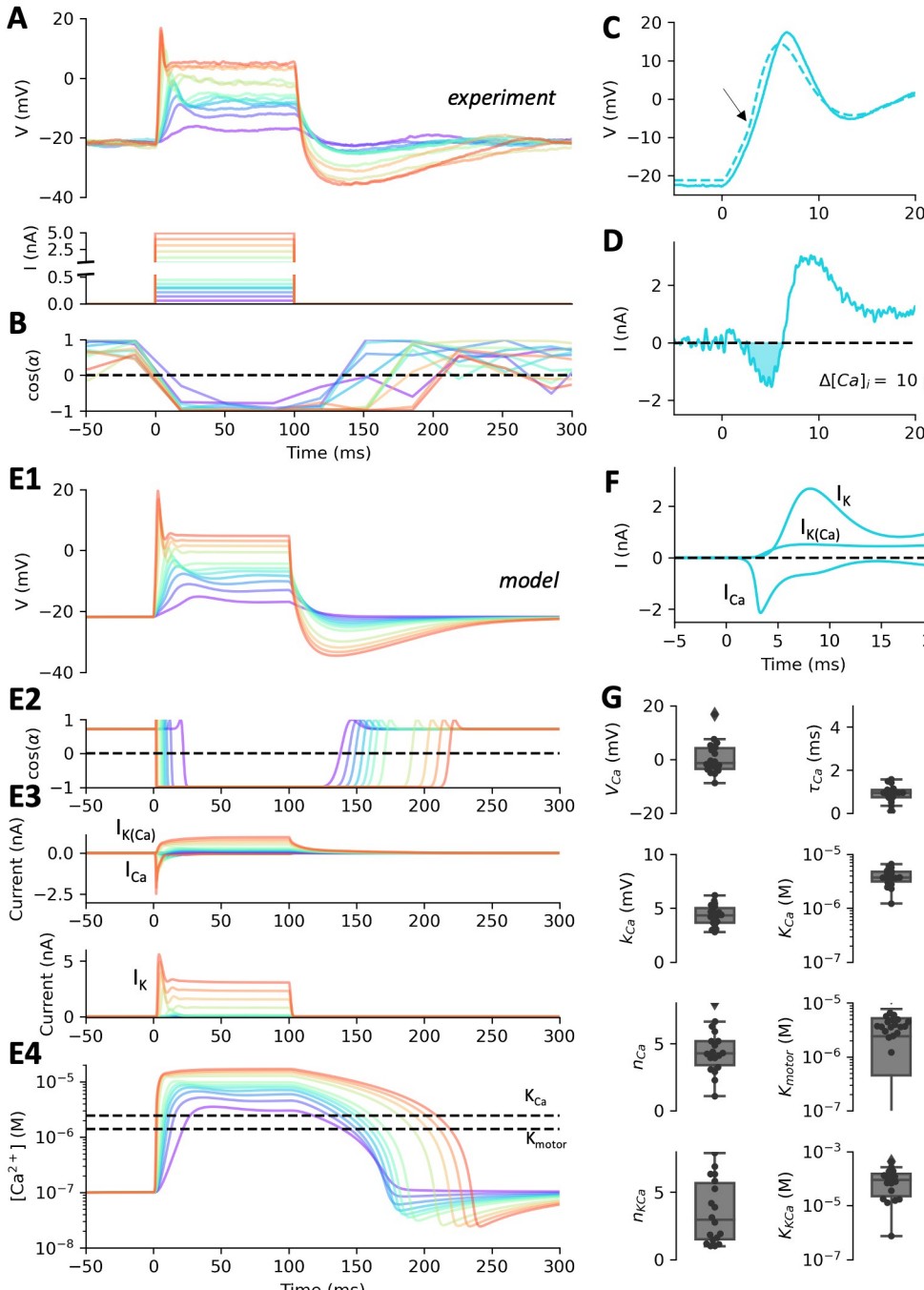

**Fig 7. Fitting the action potential of Paramecium.** A, Voltage responses of a cell (top) to two sets of current pulses (bottom), from 0 to 5 nA (in 300 pA increments) and from 0 to 300 pA (in 25 pA increments). B, Ciliary response to the same currents, measured as the cosine of the mean angle of the velocity field, relative to the anteroposterior axis. C, Close up of an action potential triggered by a 1.5 nA current pulse, with the model fit (dashed). The arrow points at an upward deflection due to the calcium current. D, Ionic current calculated by subtracting the estimated leak current from the capacitive current. The inward current (I<0, shaded) corresponds to the calcium current. Integrating this current yields a calcium entry corresponding to a 10 µM increase in intraciliary calcium concentration. E, Responses of the fitted model. E1, Voltage responses. E2, Ciliary responses. E3, Voltage-gated calcium current $I_{Ca}$ (top, negative traces), delayed rectifier K⁺ current $I_{Kd}$ (bottom) and calcium-activated K⁺ current $I_{K(Ca)}$ (top, positive traces) in the fitted model. Currents are shown with the electrophysiological convention, i.e., I<0 is inward. E4, Intraciliary calcium concentration in the fitted model. The dashed lines show the ciliary reversal threshold and the half-inactivation concentration. F, Ionic currents inferred by the model for the action potential shown in C. G, Statistics of fitted parameters (n = 18).

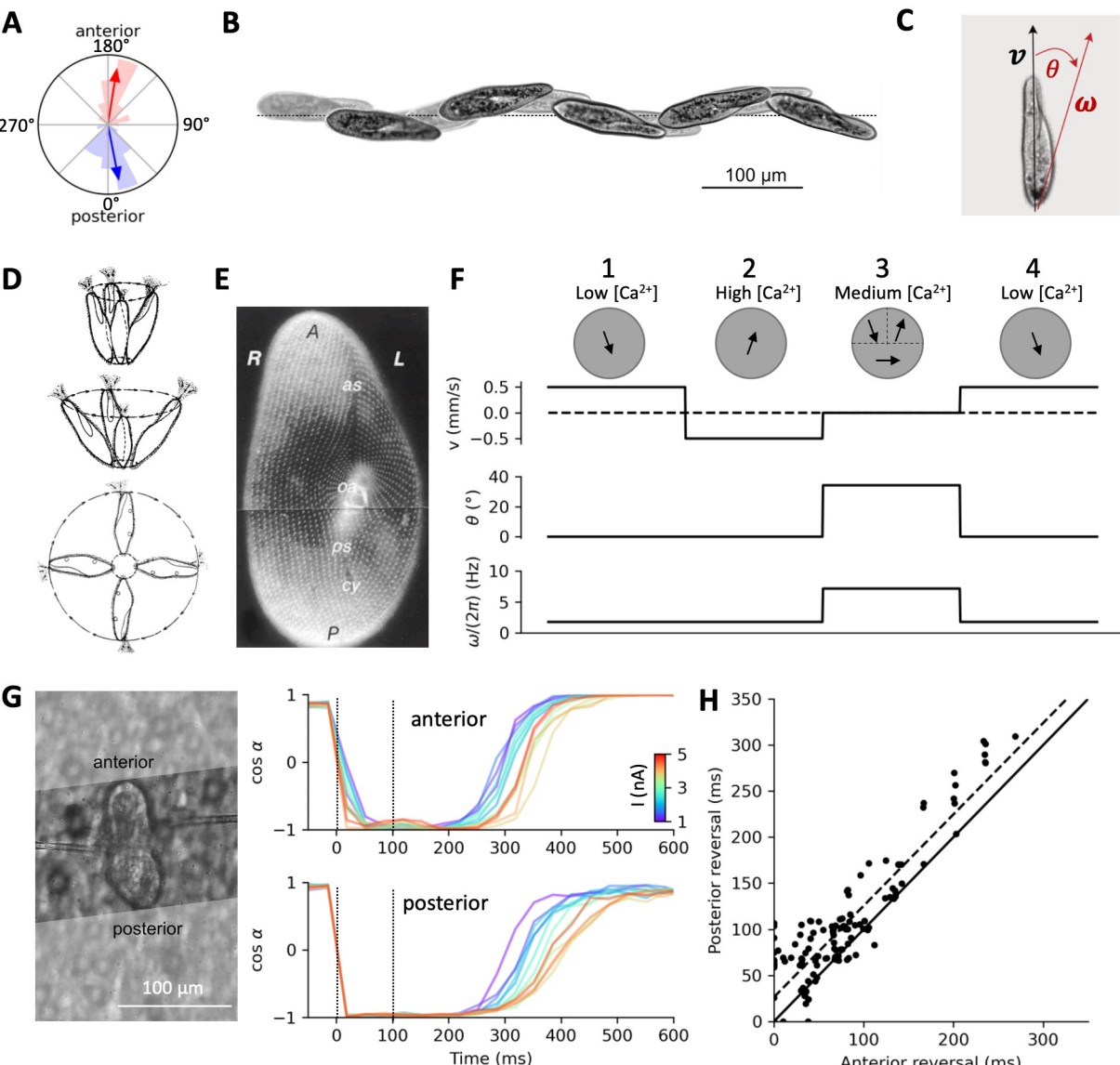

**Fig 8. Swimming and turning.** A, Direction of fluid motion during forward swimming (blue) and backward swimming (red), relative to the anteroposterior axis. Averages are shown by arrows. B, Example of helicoidal motion of Paramecium, with the oral groove facing the axis. Highlighted frames are spaced by 750 ms. C, The translational velocity vector **v** is oriented along the anteroposterior axis. The rotation vector **ω** is in the dorsoventral plane (including the oral groove), making an angle θ with the anteroposterior axis. D, Rotating movement of the cell at the end of avoiding reactions of increasing strength [27]. E, Organization of ciliary basal bodies on the oral (ventral) side [10]. The oral apparatus (oa) is in the center (R: right; L: left; A: anterior; P: posterior; as: anterior suture; ps: posterior suture; cy: cytoproct). F, Calculation of kinematic parameters v, θ and ω in a spherical model of radius 60 μm, during successive phases of the avoiding reaction. First column: cilia beat to the rear and right, producing an axisymmetric force field pushing the organism forward while spinning around its axis. Local force amplitude is adjusted for a velocity of 500 μm/s. Second column: cilia revert and now beat to the front and right, pushing the organism backward. Third column: anterior left cilia revert back to the initial direction while anterior right cilia still beat towards the front, and posterior cilia partially revert, beating to the right. Translational velocity is now 0 and the rotation axis tilts to about 34˚. Spinning speed ω also increases by a factor four. Fourth column: all cilia revert back to the initial beating direction. G, Measurement of fluid velocity in a sample cell beyond the anterior end (top) and beyond the posterior end (bottom), in response to 100 ms positive current pulses (1–5 nA), relative to the anteroposterior axis. H, Over n = 9 cells, the direction of posterior motion reverts back about 30 ms after anterior fluid motion (dashed line: linear regression; solid line: identity). Reversal duration is calculated as the time when cos(α) crosses 0, relative to the pulse end time.

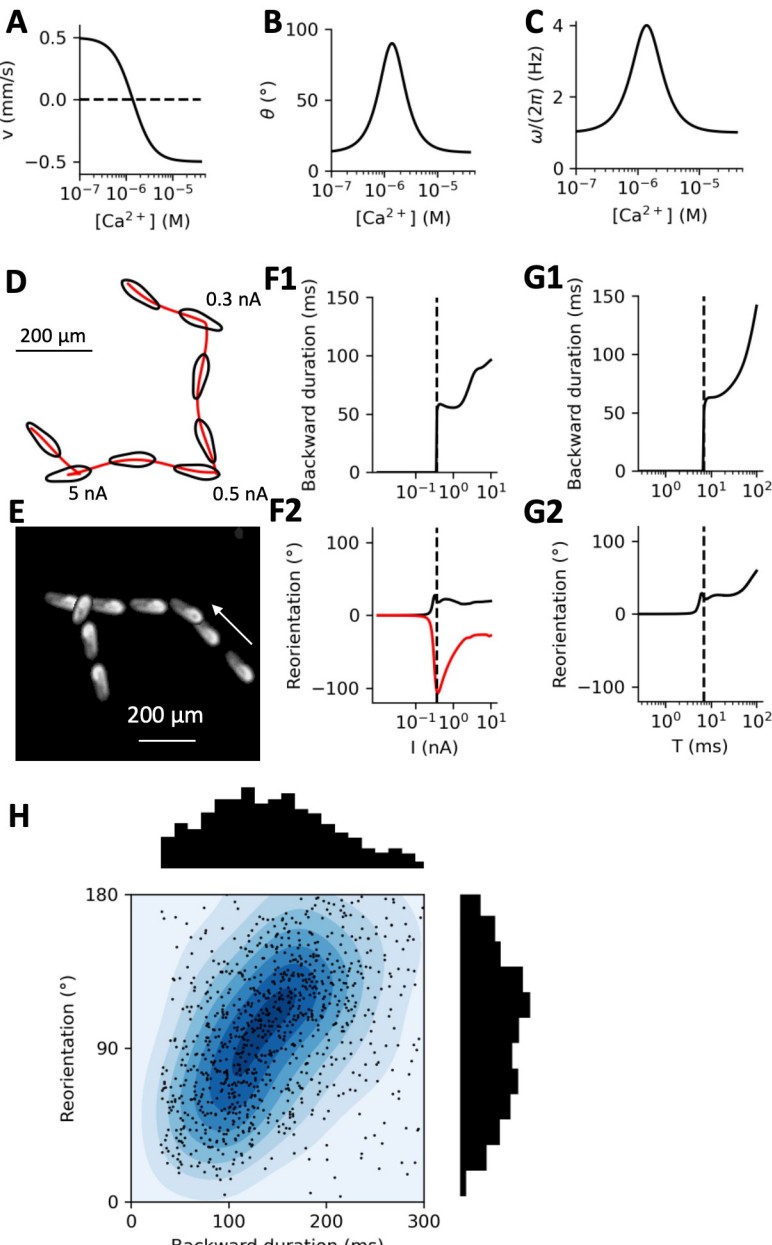

**Fig 9. Simulation of the avoiding reaction.** A, Velocity v as a function of intraciliary calcium concentration $[Ca^{2+}]$ in the model. B, Angle $\theta$ of the rotation axis as a function of $[Ca^{2+}]$ in the model. C, Spinning speed $\omega$ as a function of $[Ca^{2+}]$ in the model. Angle and spinning speed increase at intermediate $Ca^{2+}$ concentration, as implied by the spherical model in Fig 8F and 8D, Simulated model trajectory with three 2 ms current pulse stimulations of increasing amplitude. Images are shown at 400 ms intervals. Without stimulation, the organism swims in spiral, with the oral groove facing the spiral axis. A very small stimulation deviates the trajectory. Stronger stimulations produce avoiding reactions, with backward swimming and turning. E, Example of an observed Paramecium trajectory showing a directional change without backward swimming (right), followed by a full avoiding reaction (left). Images are shown at 400 ms intervals, starting on the right. F, Backward swimming duration (F1) and reorientation angle (F2) as a function of current amplitude for 2 ms pulses. Red and black curves show results for the same model but different initial positions of the oral groove, differing by a quarter of a cycle. G, Backward swimming duration (G1) and reorientation angle (G2) as a function of current pulse duration T with 100 pA amplitude. H, Reorientation angle vs. backward swimming duration in n = 1138 spontaneous avoiding reactions of Paramecium, showing a positive correlation (linear regression r = 0.2, p ≈ $10^{-11}$). About 15% of data points are not represented (larger angle or duration). Colors represent contour lines of the distribution.

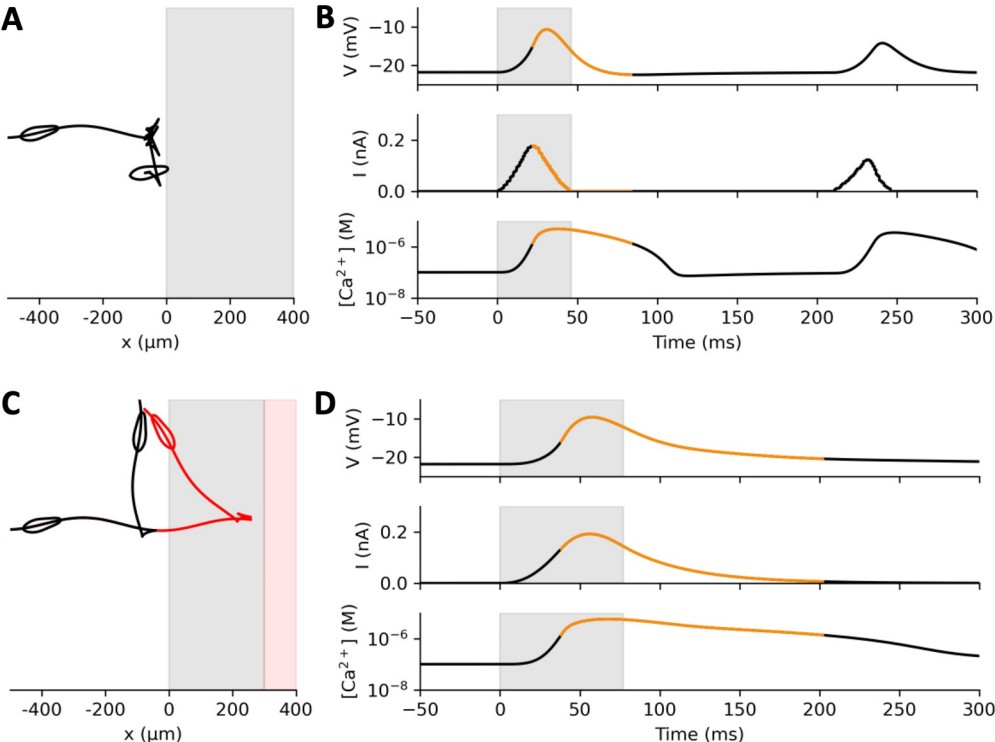

**Fig 10. Interaction of a model Paramecium with a generic stimulus, modelled as a positive current proportional to the cell area within the stimulus area.** A, Trajectory of the model doing several avoiding reactions against the stimulus. B, Membrane potential (top), stimulus current (middle) and intraciliary calcium concentration (bottom) at the first contact. Contact occurs at the boundary with the shaded region. Orange curves indicate backward swimming. Several weak avoiding reactions occur in succession. C, Trajectory of the model where sensory transduction has a 40 ms activation/deactivation time constant. In red, the stimulus is placed 300 μm further away. D, Same as B, for the black trajectory in C. The stimulus current lasts longer and peaks after the organism has started reacting, resulting in a stronger avoiding reaction.

depolarization, and the inward rectifier current $I_{Kir}$, activated upon hyperpolarization. These two currents are indeed observed in electrophysiological measurements (Fig 6A).

We fitted a Boltzmann model of the delayed rectifier current (see Methods, *Electrophysiological modeling*, Eqs (6), (7), (12) and (13)), $I_{Kd} = g_{Kd}n^2(E_K - V)$ ($g_{Kd}$ is the maximal conductance, $n$ is the gating variable), to responses to 100 ms depolarizing current pulses (0 to 4 nA in 300 pA increments). This model turned out to fit the data as well as a Hodgkin-Huxley model, but with fewer parameters (see Methods, *Model optimization*). Fig 6B (bottom) shows responses of this model, and Fig 6C shows the activation curve and voltage-dependent time constant with median parameters and those for the cell shown in Fig 6B, with detailed statistics in Fig 6D. The delayed rectifier current activates at a median value of $V_{Kd}$ ≈ 21 mV (30 ± 23 mV) with a slope $k_{Kd}$ ≈ 7 mV (9 ± 6 mV). The time constant peaks at ~4.1 ms (4 ± 1.3 ms) at a voltage $V_\tau$ ≈ 23 mV (26 ± 48 mV), with a slope $k_\tau$ ≈ 12 mV (14 ± 8 mV).

Based on these results, we further simplified the model by enforcing $V_{Kd} = V_\tau$ and $k_\tau = 2k_{Kd}$. This simplification slightly increases the fit error (1.82 vs. 1.8 mV; p = 0.009, two-tailed Wilcoxon test), but reduces the number of parameters. We used this simplified model in the full model of ciliated cells (leaving its parameters unconstrained).

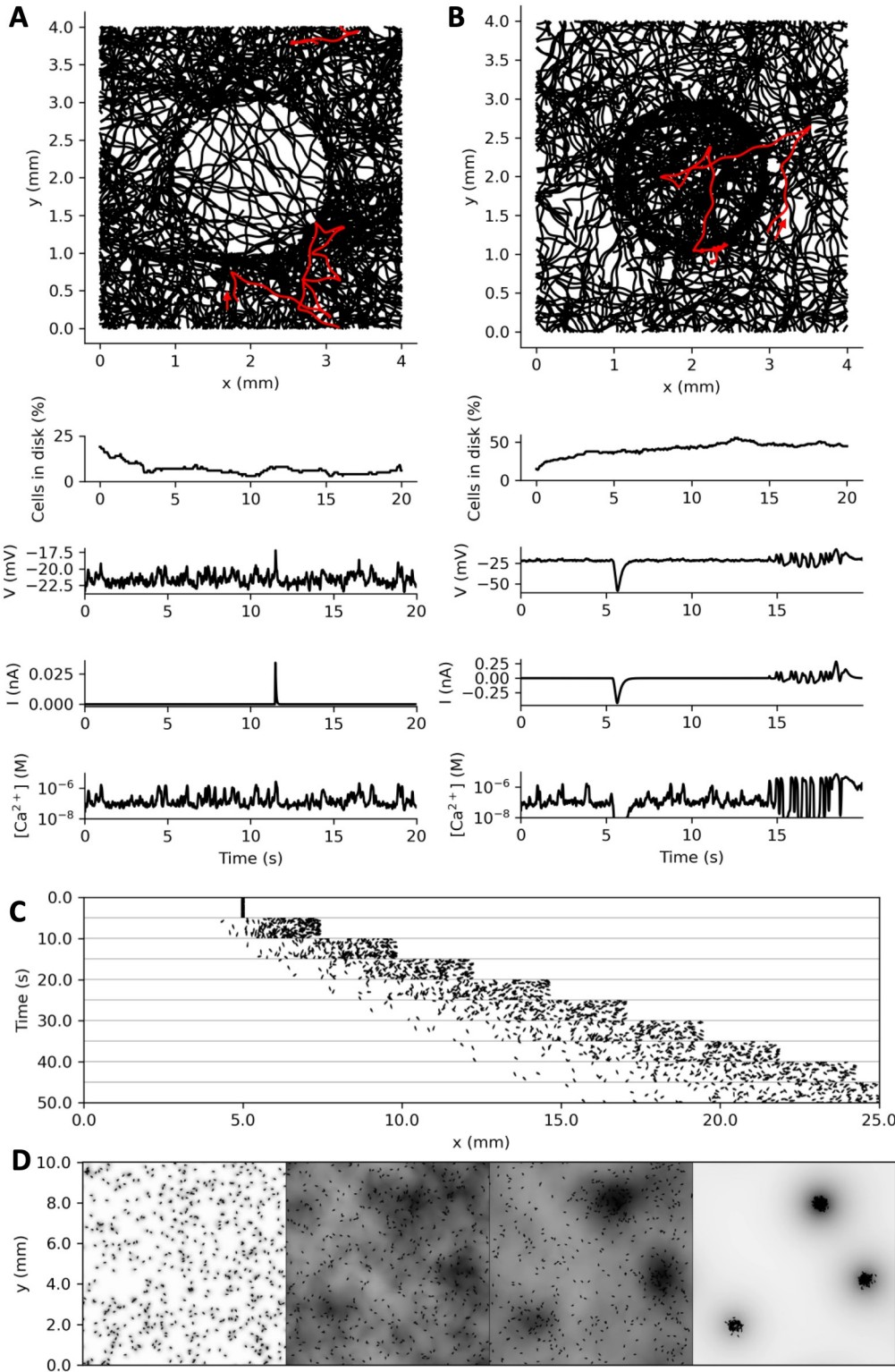

**Fig 11. Closed-loop behavior of the Paramecium model.** A, Top: trajectories of 100 models swimming for 20 s in a torus with a depolarizing circular stimulus, modelled as in Fig 10C. The proportion of cells in the disk quickly decays (below). Membrane potential, stimulus current, and intraciliary calcium concentration are shown for the trajectory highlighted in red, which does an avoiding reaction against the disk after a number of spontaneous avoiding reactions. B, 100 model trajectories with a circular stimulus triggering an adapting hyperpolarizing current. Organisms tend to

make avoiding reactions on the inner boundary of the disk. The proportion of cells in the disk increases over time. The highlighted trajectory enters the disk around t = 5 s with a large hyperpolarization, then displays several avoiding reactions against the boundary of the disk before exiting. C, Paramecia swimming in a linear stimulus gradient, modelled as in B. The position of 200 cells starting at position x = 5 mm is displayed every 5 s. D, Collective behavior in model paramecia induced by respiration and chemosensitivity. $CO_2$ produced by cells is displayed in shades of grey (normalized to the spatial peak), and diffusion is simulated. $CO_2$ concentration represents an attracting stimulus modelled as in B and C.

## The action potential

We now build a model of the action potential of ciliated cells, coupled with cilia reversal (Fig 7A). Cilia revert with very small depolarizations, of just a few mV [42]. For this reason, we used two sets of pulses, large pulses from 0 to 5 nA in 300 pA increments, and small pulses from -100 to 500 pA in 25 pA increments (Fig 7A, bottom; only positive currents are shown). Tip potentials could fluctuate between these two sets, therefore we aligned the traces to the median resting potential of -22 mV, and we fixed $E_K$ at its median value of -48 mV. Simultaneously, we seeded the extracellular medium with 1 μm tracer particles and imaged their motion at a frame rate of 30 Hz. Particle image velocimetry was then used to calculate the fluid velocity field, giving an indication of the direction of ciliary beating, as illustrated in Fig 2C (See Methods, *Particle image velocimetry*). Fig 7B shows the cosine of the mean angle $\alpha$ of the velocity field during stimulation, relative to the cell's anteroposterior axis: 1 means that particles flow towards the posterior end, i.e., the cell is trying to swim forward; -1 means that the cell is trying to swim backward. Thus, cilia revert for a duration longer than the pulse (120–200 ms in this cell), graded with pulse intensity.

In the electrophysiological responses, we notice a small upward deflection after stimulation as shown in Fig 7C (arrow). This deflection is due to an inward current, the $Ca^{2+}$ current. This current can be estimated by subtracting the estimated leak current from the capacitive current $C.dV/dt$ (passive properties estimated by model fitting, see below). With a pulse of intensity $I$ = 1.5 nA, we find that the inward part of that current peaks at about 2 nA (Fig 7D). Note that in this Figure, currents are shown with the electrophysiological convention so as to compare with previous voltage clamp studies [18], i.e., I<0 is inward; this is the opposite of the modeling convention. The peak amplitude shown in Fig 7D is an underestimation of the calcium current since part of the inward current may be masked by the $K^+$ current, but it is comparable to previous estimations in voltage-clamp [25]. The calcium current is known to activate and inactivate quickly, within a few ms [25,31], as can be seen on Fig 7D.

We also observe small oscillations (Fig 7A), due to the interplay between $Ca^{2+}$ and $K^+$ currents, and a pronounced hyperpolarization after the pulse. This hyperpolarization is due to a calcium-activated $K^+$ current $I_{K(Ca)}$. This current has been previously characterized electrophysiologically [35,36], as well as genetically and with immunochemistry [12,34].

Thus, we included the following currents in our model: a leak current $I_L$, a voltage-gated calcium current $I_{Ca}$, with calcium-mediated inactivation, a delayed rectifier $K^+$ current $I_{Kd}$, and a calcium-activated $K^+$ current $I_{K(Ca)}$ (see Methods, *Electrophysiological modeling*). The calcium current $I_{Ca}$ is produced by ciliary channels similar to L-type Cav1.2 channels [16]. We modeled it similarly to [32,53], but we allow for several inactivation binding sites (Eqs (14)–(16)). In addition, the current uses the Goldman-Hodgkin-Katz equation, which is more appropriate than the linear driving force ($E_{Ca}$—V) when intra- and extracellular concentrations are very different [54]. The calcium-activated $K^+$ current $I_{K(Ca)}$ is simply modelled with a conductance increasing as a Hill function of calcium concentration (Eqs (17) and (18)).

In addition, the model must include calcium dynamics (Eq (19)). The decay of calcium concentration after the action potential may be due to a combination of processes, including diffusion towards the base, buffering (in particular by centrin), and pumps. We model this combination by a simple linear model of the calcium flux. However, this is not sufficient because there is a large calcium flux at rest through the voltage-gated calcium channels, which must also be expelled or buffered. Thus, we postulate that the resting calcium concentration, about 0.1 μM [55,56], is maintained by a pump operating near that concentration, which is consistent with properties of plasma membrane calcium pumps (PMCA), also present in the cilia [33,34]. We model this pump with Michaelis-Menten kinetics.

Finally, we couple calcium concentration with ciliary beating angle by a Hill function (Eq (20)).

We then fitted this complete model simultaneously to electrophysiological and motor responses to 100 ms current pulses (n = 18), while ensuring that the resting calcium concentration was 0.1 μM (Fig 7E). Thus, calcium concentration is not directly measured, but indirectly constrained by several processes: inactivation of $I_{Ca}$, activation of $I_{K(Ca)}$, and ciliary reversal.

Fig 7E1 and 7E2 show the fits for the cell shown in Fig 7A and 7B. Consistently with previous voltage-clamp measurements [25], the inferred calcium current is transient and peaks at about ~2.5 nA (-2.5 nA with the electrophysiology convention, Fig 7E3). A residual current remains, so that calcium concentration remains high in the model during stimulation (Fig 7E4). This is consistent with the fact that ciliary reversal can last for many seconds when the membrane is depolarized [42]. With large currents, inferred calcium concentration raises to about 22 μM, similar to previous estimations.

The voltage-gated potassium current is delayed relative to the calcium current, and the calcium-activated K$^+$ current raises more slowly and is only dominant during repolarization (Fig 7E3), with a maximum of 0.65 nA. This is consistent with previous studies of that current [36]. Fig 7F shows the three different currents during the action potential shown in Fig 7C. As previously argued, the calcium-activated K$^+$ current has a small contribution to the early current [25].

Over the *n* = 18 cells, we find in the fitted models that the calcium current has half-activation voltage $V_{Ca}$ = -1 mV (0 ± 6 mV), activation slope *k* = 4.3 mV (4.3 ± 1 mV) and time constant about 0.9 ms (0.9 ± 0.4 ms) (Fig 7G). Estimated conductance is not well constrained and often very large. This is presumably because the peak current is mainly determined by the inactivation properties, and therefore the conductance parameter is not well constrained. Half-inactivation occurs at about $K_{Ca}$ = 3.7 μM (log$_{10}$(K$_{Ca}$ in M) = -5.4 ± 0.2). This is close to patch-clamp measurements on cardiac L-type calcium channels [57]. The fitted models have about 4 binding sites (4.4 ± 1.7), larger than previous models [32,53] (which have a single site but were not constrained by *Paramecium* data). Calcium decays in the model with a median time constant of 130 ms and the pump operating near rest has a median maximum rate of 87 μM/s.

The delayed rectifier current has similar fitted parameters as in deciliated cells (median $k_{Kd}$ = 4.9 mV, $\tau_{Kd}$ = 4.5 ms), except half-activation tends to be lower (median $V_{Kd}$ = 4 mV)–this might be because the responses are essentially below V$_{Kd}$, in the unsaturated part of the activation curve. The calcium-activated K$^+$ current $I_{K(Ca)}$ has low affinity ($K_{KCa}$ = 63 μM, log$_{10}$(K$_{KCa}$ in M) = -4.2 ± 0.7). This is consistent with the observation that in voltage-clamp, this current keeps on increasing for at least one second [36]. There are about n$_{KCa}$ = 3 binding sites is (3.5 ± 2.3).

Finally, cilia revert when the inferred calcium concentration is about 2.4 μM (log$_{10}$(K$_{motor}$ in M) = -5.9 ± 0.8). This is close to measurements with triton-permeabilized cells, reporting about 1 μM [30]. We note that this and other concentration parameters depend on the estimation of intraciliary volume, which is approximate.

Overall, parameters of the fitted models are compatible with known properties of the currents and of ciliary reversal.

## Swimming and turning

We now examine how *Paramecium* swims and turns, before coupling the electrophysiological model with swimming motion.

Before stimulation, the flow produced by the cilia is directed towards the posterior end, about 11˚ to the right (Fig 8A, blue). This should produce a forward left spiraling movement, as documented from observations of free swimming [27,28]. During a pulse that triggers an action potential, the flow is directed towards the anterior end, about 9˚ to the right (Fig 8A, red). This would make the cell swim backward, also spiraling to the left. An example of this spiraling motion is shown on Fig 8B: the organism swims forward while spinning around a tilted axis. The oral groove faces the spiral axis [13,29], which means that the rotational velocity vector $\omega$ is tilted from the main axis towards the oral side by an angle $\theta$ (in the median plane; Fig 8C). In freely swimming paramecia, we found that $\theta \approx 13˚$ ($\pm 6.4˚$) and the rotation speed $\|\omega\|$ is about 1 cycle/s ($1.03 \pm 0.2$ cycle/s) (see Methods, *Behavioral measurements* and *Behavioral analysis*).

How does *Paramecium* turn? A directional change can occur if the angle $\theta$ changes. According to Jennings [27], $\theta$ increases during the avoiding reaction, in relation with stimulus strength (Fig 8D). In the final model, we will directly couple the calcium concentration [$Ca^{2+}$] to the three kinematic variables v, $\theta$ and $\omega$. In order to postulate plausible calcium-kinematic couplings, and to understand the relationship between ciliary beating patterns and kinematic parameters, in particular $\theta$, we now examine an idealized hydrodynamic model, consisting of a sphere of radius 60 μm, for which we can use analytical formula relating forces and motion (see Methods, *Hydrodynamic model*). Fig 8E shows the ciliature on the ventral (oral) side, which appears to be organized around the oral groove. In the model, we postulate that cilia on the left (L), right (R), anterior (A) and posterior (P) sides may beat in different directions. The fluid produces local forces opposite to the direction of ciliary beating, and at low Reynolds number, the total force and torque map linearly to the translational velocity vector **v** and the rotational velocity vector $\omega$ in the cell coordinate system [58].

At rest (low calcium concentration), cilia beat towards the rear, slightly to the right (Fig 8F, first column), so that the fluid produces a force towards the front, slightly to the left. If the direction of ciliary beating is identical everywhere in spherical coordinates (that is, in terms of the cardinal directions North/South/East/West, Fig 12), then the force field over the sphere is symmetrical with respect to the main axis. This makes both the total force and the total torque align with the main (antero-posterior) axis, and therefore **v** and $\omega$ are also aligned with that axis, that is, $\theta = 0$. The organism then moves forward, with a spinning movement around the axis. We adjust the force so that the velocity is 500 μm/s, which makes the sphere spin at about 1.8 Hz. Upon stimulation, when calcium concentration is high, cilia revert and beat forward (Fig 8F, second column), making the organism move backward.

Thus, the organism cannot turn unless there is some asymmetry in the ciliary beating pattern. Machemer [59] and Párducz [60] observed that during the turning phase, anterior and posterior cilia beat in different directions; Jennings [27] observed that left and right anterior cilia beat in different directions, where "left" and "right" are relative to the oral groove. In Fig 8F (third column), we examine what happens if cilia beat in a swirling pattern around the oral groove: the left anterior cilia beating backward, the right anterior cilia beating forward, and the posterior cilia beating to the right. This corresponds to what would happen near the calcium concentration threshold for global ciliary reversal, if cilia revert back first in the left

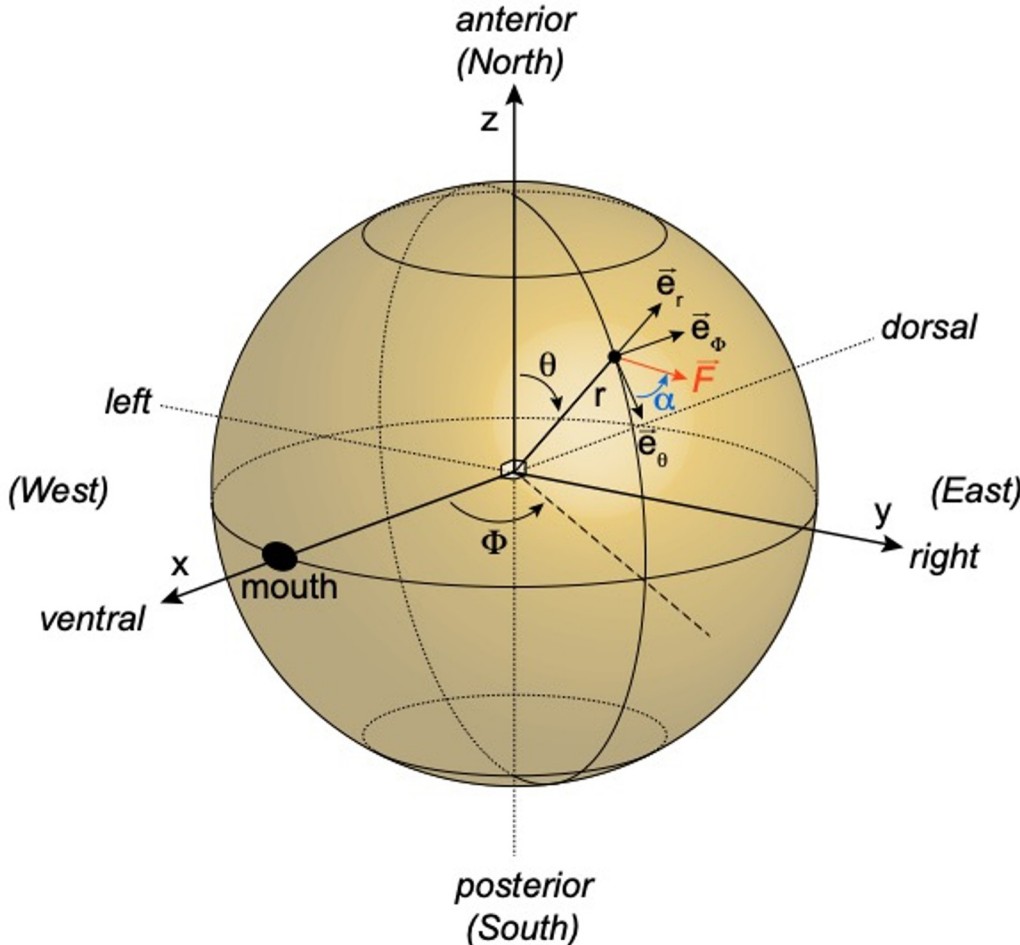

**Fig 12. Conventions on the spherical model, with spherical coordinates and a local force F on the surface of the sphere.**

anterior part, then in the posterior part, then in the right anterior part. The swirling pattern suggests that the cell is going to turn around an axis tilted from the main axis, in the plane separating the left and right sides, and this can be confirmed analytically. In this configuration, the sphere does not move along the main axis ($v = 0$ mm/s), because the net force along that axis is null, but it turns along an axis tilted by $\theta \approx 34°$ from the main axis. The sphere also spins about 4 times faster ($\omega/(2\pi) \approx 7.2$ Hz)–this is of course qualitative since *Paramecium* is not a sphere. It returns to moving forward when all cilia revert back (Fig 8F, fourth column).

This pattern of ciliary reversal is suggested on Fig 2C. However, given that the particle flow was measured on a plane ~30 μm above the cell, that the cell could take different shapes and that the position of the oral groove was often difficult to estimate, it was generally not possible to determine the precise pattern of ciliary reversal empirically. Nevertheless, it is possible to demonstrate that cilia revert asynchronously. We measured particle flow separately in two regions of the field, beyond the anterior end, and beyond the posterior end (Fig 8G). This was only possible for 9 cells, where video quality was sufficient (see Methods, *Particle image velocimetry*). We measured the mean angle $\alpha$ of the flow field during stimulation with current pulses between 1 nA and 5 nA; weak pulses were not included because particle density tended to be lower due to sedimentation (weak pulses were recorded after strong pulses).

Fig 8G shows in one cell that posterior cilia (bottom) revert back after anterior cilia (top). Thus, before stimulation, when calcium concentration is low, anterior and posterior cilia beat in the same direction. Beating direction is also spatially homogeneous during stimulation, when calcium concentration is high. However, after stimulation, anterior and posterior cilia beat in different directions for a short period. This was a reproducible finding across the 9 cells: the posterior side tends to revert back slightly later than the anterior side (Fig 8H) (p = $7.10^{-17}$, one-tailed Wilcoxon test), with a mean delay of 26 ms (s.d. 29 ms). This confirms Párducz' observations [60], which were based on electron microscopy of cells fixed during the avoiding reaction.

## Modeling the avoiding reaction

This idealized hydrodynamic model shows that an asynchronous control of cilia across the membrane by calcium can make the organism turn, by transiently increasing the rotation angle $\theta$ and the spinning speed $\omega$. The exact mechanism by which calcium might differentially activate the cilia is beyond the scope of this paper (see Discussion). Here we simply postulate phenomenological couplings between calcium concentration and kinematic parameters, as suggested by our analysis (note that the hydrodynamic model is not explicitly included in our final model, as it was only used to explore the relation between ciliary patterns and kinematics). As illustrated in Fig 8C, we will assume that the translational velocity vector **v** is aligned with the main axis, so that it is fully parameterized by the velocity $v$, and that the rotation vector $\boldsymbol{\omega}$ lies in the plane of the oral groove, so that it is parameterized by its angle $\theta$ relative to the main axis and the spinning speed $\omega$. We assume that all three kinematic parameters ($v, \theta, \omega$) are functions of intraciliary calcium concentration $[Ca^{2+}]$ Fig 7A–7C) (See Methods, *Electromotor coupling*, Eqs (21)–(23)).

We model velocity as a Hill function of $[Ca^{2+}]$, with threshold equal to the reversal threshold $K_{Ca}$ and $n = 2$ sites, linearly scaled to match the maximum positive and negative velocities measured empirically Fig 7A; (Eq (21)). In freely swimming paramecia, we observed that the median velocity was 472 μm/s (521 ± 285 μm/s) for forward swimming and 370 μm/s (411 ± 200 μm/s) for backward swimming. Thus, in the model, we simply set both forward and backward maximum velocity at $v = \pm500$ μm/s. Fig 9A shows the resulting function for the cell shown in Fig 7A–7F.

We model both $\theta$ and $\omega$ as bell functions of $[Ca^{2+}]$ Fig 9B and 9C; Eqs (22) and (23); see Methods, *Electromotor coupling*), peaking when $[Ca^{2+}]$ is near the global ciliary reversal threshold $K_{motor}$ (2.4 μM), as suggested by our analysis of the spherical model (Fig 8F). The minimum angle is taken from measurements of trajectories of freely swimming paramecia ($\theta \approx 13°$). For the maximum angle, we choose $\theta = 90°$ to account for the strongest avoiding reactions observed by Jennings (Fig 8D), corresponding to a rotation normal to the main axis. The minimum spinning speed is based on measurements ($\omega \approx 1$ cycle/s), and the maximum spinning speed is set to 4 times the minimum, as in the spherical model (Fig 8D).

In this way, we obtain a model in which all kinematic variables are coupled to the electrophysiological model. We note that, in contrast with the electrophysiological part, this model is only loosely constrained by measurements (see Discussion). We can then calculate organism motion from these variables, and thereby simulate behavior in an environment (see Methods, *Kinematics*). In the following, the model of one particular cell is chosen for illustration, the same cell as in Fig 7.

We first examine the trajectory of a model stimulated by 2 ms currents of varying amplitude (Fig 9D) (see Methods, *Behavioral scenarios*). Without stimulation, the organism swims in a

helicoidal path. With a small stimulation amplitude (0.3 nA), the organism changes direction without swimming backward. At larger amplitude (0.5 nA), the organism swims backward for a very short time then turns and swims forward. When the amplitude is increased (5 nA), backward swimming is more noticeable. Directional changes without backward swimming do occur in freely swimming paramecia, as illustrated on Fig 9E: the organism first changes direction without swimming backward, then does an avoiding reaction.

In more detail (Fig 9F1), we observe that the cell swims backward when current intensity exceeds a threshold (here 372 pA), then backward swimming duration tends to increase with intensity. The reorientation angle following the stimulation changes continuously with stimulation strength, but in a complex way (Fig 9F2). In particular, small stimulations can trigger large turns without backward swimming. In addition, the directional change depends on the initial position of the oral groove: the red and black curves of Fig 9F2 correspond to the same cell but an oral groove position (spinning angle) differing by a quarter of a cycle. This occurs because the observation plane is fixed while the organism spins around its main axis, and because movement is constrained in the plane.

The characteristics of the avoiding reaction also depend on stimulus duration (Fig 9G). If the pulse amplitude is fixed (I = 0.1 nA) and its duration is increased, then the duration of backward swimming increases (Fig 9G1), and reorientation angle tends to increase for large durations but is not monotonous near threshold (Fig 9G2).

When we examine spontaneous avoiding reactions of freely swimming paramecia, we find that both backward swimming duration and reorientation angle vary broadly ($156 \pm 81$ ms and $114 \pm 66°$, respectively) (Fig 9H), and there is a small although highly significant correlation (linear regression, $r = 0.2$, $p = 10^{-11}$). Thus, backward swimming duration and reorientation angle are variable and not deterministically related.

## A closed-loop behavioral model of *Paramecium*

We now use the model in closed loop, to describe how the interaction between organism and environment may give rise to behavior (see Methods, *Behavioral scenarios*), as opposed to motor responses to predetermined input stimuli (open loop). The following scenarios are relatively abstract and are meant mainly as illustrations of the possible uses of this integrative model to investigate the relation between physiology and behavior.

First, we consider an organism swimming towards a generic object, which triggers a depolarizing current when in contact with the membrane (for example a chemical substance, or hot water) (Fig 10). Thus, we simply consider that the stimulus current is proportional to the surface area in contact with the stimulus (see Methods, *Sensory transduction*). In contrast with previous situations, the stimulus is not pre-determined but depends on behavior. As Dewey pointed out (1896), "*the motor response determines the stimulus, just as truly as sensory stimulus determines the movement.*". When the organism touches the object, a current is triggered, which depolarizes the membrane (Fig 10A and 10B). As the cell swims into the object, the current increases until an action potential is triggered. The cell then swims backward, moves out from the object and the current stops. Thus, the sensory current is necessarily small and short, because the organism withdraws as soon as the current reaches threshold. This results in a small avoiding reaction, and the organism bumps again repetitively against the object until it finally escapes (S1 Movie).

Larger movements can be obtained if sensory transduction has slower kinetics Fig 10C and 10D and S2 Movie). Here, the sensory current follows the stimulation with a time constant of 40 ms, modelled with first order kinetics (to simplify, the spatial spread of channels is not modeled; (Eq (24)). In this case, the sensory current keeps on increasing (slightly) after the

organism has started swimming backward and it lasts longer. This results in a larger avoiding reaction.

Although the model is deterministic, the directional change can be described as pseudo-random. If the object is moved 300 µm further (Fig 10C), then the organism escapes with a larger angle. This is because the organism spins while swimming, so that its oral groove takes a different position when it touches the object.

We now consider that the object is a disk in a square environment (Fig 11A and S3 Movie). To avoid boundary effects, we consider that the environment has the topology of a torus (paramecia escaping to the left reappear to the right). To account for spontaneous avoiding reactions (occurring at a rate of about 0.2 Hz in our behavioral measurements), we added a noisy current to the membrane equation. The paramecia are modeled as in Fig 10C and 10D, with slow transduction. Fig 11A shows 100 trajectories starting from random positions and simulated for 20 s. The proportion of paramecia inside the disk decreases from 19 to 7%. Thus, the disk acts as a repelling stimulus. Occasionally, a paramecium gives an avoiding reaction against the boundary, makes a large turn and swims backward into the disk. In this case, it continues being stimulated and swims backward through the disk until it escapes. This peculiar behavior might be avoided if currents of opposite polarity were triggered when stimulating the rear, as is the case for thermal stimuli [61] and some chemical substances [62]. Traces shown in Fig 11A show the membrane potential $V$, the stimulus current $I$ and the calcium concentration $[Ca^{2+}]$ for the trajectory shown in red, where we can distinguish a number of spontaneous avoiding reactions and one avoiding reaction against the disk.

Next, we examine how an object can act as an attractive stimulus Fig 11B and S4 Movie). This behavior can be obtained if the cell responds to a hyperpolarizing stimulus with adaptation, so that a depolarization is triggered when the stimulus stops. A simple model that exhibits this behavior is one where the stimulus triggers currents through two pathways with different kinetics: a hyperpolarizing current with fast kinetics, and a depolarizing current of equal magnitude with slow kinetics (Eqs (25)–(27)). In this way, a transient hyperpolarizing current is triggered when the stimulus switches on, and a transient depolarizing current is triggered when the stimulus switches off. This is shown on Fig 11B. The proportion of paramecia inside the disk increases with time (from 15% to 45% in 20 s). It can be seen that paramecia tend to aggregate inside the disk, mostly near the boundary. This is due to the small avoiding reactions but also to the curvature of the disk.

Using the same model, we then place the paramecia in a linear stimulus gradient (Fig 11C and S5 Movie); the environment is toric in the transversal dimension (up and down boundaries are glued). We place all paramecia at the same initial position, with random orientations. The population then rapidly ascends the gradient. This occurs because a depolarizing current is produced when the stimulus decreases, triggering an avoiding reaction. This behavior has been observed in *Paramecium* with thermal gradients [63,64], and shares similarities with bacterial chemotaxis [65].

Finally, we show an example of collective behavior (Fig 11D and S6 Movie). *Paramecium* produces $CO_2$ by respiration, which acidifies its medium, and it is attracted by weak acids [66–68]. As a result, it can form aggregates, for example around a source of food or at the bottom of a depression slide [68]. We simulate the production of $CO_2$ by paramecia and its diffusion in the medium (a square with torus topology), together with sensitivity to $CO_2$ modeled in the same way as in Fig 9B and 9C (see Methods, *Collective behavior*). $CO_2$ concentration is represented in Fig 11D as grey shades, after normalization. In this simulation, paramecia progressively form aggregates.

## Discussion

### Summary

We have built an integrative model of *Paramecium* that combines electrophysiology and motility. The model is informed by previous experimental literature and constrained by specific electrophysiology and trajectory measurements. It consists of an empirically constrained biophysical model of the action potential, with a coupling to kinematic parameters, which is more phenomenological. It can be simulated as a model of autonomous behavior in various environments.

The electrophysiological model was built by model fitting to current clamp data, with calcium-dependent properties indirectly constrained by ciliary reversal data. This method recovered properties of individual currents compatible with previous measurements obtained by different means. For example, the calcium threshold for ciliary reversal was estimated to be ~2 μM, the same order of magnitude as measured by varying extracellular calcium concentration in *Paramecium* with permeabilized membranes [30]. This is notable since calcium was not measured but only inferred from electrophysiology (indirectly through the estimation of the calcium current by the fitting procedure). The fitting procedure also determined that the calcium-dependent $K^+$ current is small during the action potential, as previously determined with voltage-clamp experiments [25], but dominant after stimulation. The magnitude and time scale of calcium currents estimated by fitting were also compatible with voltage-clamp measurements [25]. Quantitative fitting allowed us to estimate additionally the calcium inactivation threshold (~3 μM) and the number of inactivation sites (~4).

By measuring ciliary induced flows during action potentials, we found that ciliary reversal is not synchronous across the cell, confirming previous observations obtained with electron microscopy [60]. We showed with a simple hydrodynamic model that asynchronous ciliary reversal allows the organism to turn, namely if the order of ciliary reversal follows a swirling pattern around the oral groove. From these findings, we built a phenomenological model of the coupling between calcium concentration and the main kinetic parameters, constrained with trajectory measurements (speed, angle).

The integrated model shows helicoidal swimming with graded avoiding reactions, where backward duration swimming and reorientation angle increase with stimulus strength or duration. As observed in spontaneous behavior, the model can also slightly reorient without swimming backward.

Behavior of the autonomous model is more complex than stimulus-response experiments, because the relation between sensory stimulus and motor response is circular. In particular, we noticed that the interaction with an object (e.g. a chemical substance) critically depends on the properties of sensory transduction. For example, efficient avoidance of the object requires persistent stimulation, e.g. with slow sensory activation/deactivation. Furthermore, sensory adaptation to a hyperpolarizing stimulus makes the stimulus attractive. Another possibility, which is non-stimulus-specific, is that excitability adapts to voltage changes [69], for which there is some electrophysiological evidence in *Paramecium* [24]. This can allow the model to follow a stimulus gradient. Finally, collective behavior can arise if organisms are sensitive to a substance that they produce. In summary, relatively complex behavior can be generated by the interaction of this simple "swimming neuron" with its environment. We note that we have not included any specific empirical model of transduction, which would be required in any empirical investigation of behavior using our model.

## Limitations

This work has many limitations. First, ionic currents were measured simultaneously rather than in isolation, although we could isolate $I_{Kd}$ by deciliation. This was partly for technical reasons (one cannot measure $I_{K(Ca)}$ while blocking the calcium current), and partly to ensure a global fit of the entire model to the action potential. Nonetheless, current overlap may cause difficulties for model fitting. For this reason, we strived to choose the simplest models that captured the phenomenology.

A second limitation is that calcium was not directly measured. Instead, it was indirectly constrained by several observed phenomena: calcium-dependent inactivation of $I_{Ca}$, calcium-dependent activation of $I_{K(Ca)}$, and ciliary reversal. Calcium imaging has been performed previously in *Paramecium* by pressure injection of a calcium indicator, in other contexts [56,70,71]. However, it is technically challenging to perform quantitative time-resolved measurements of ciliary calcium, because the cilia represent a small fraction of the total volume (2–3%) and beat at about 20 Hz.

Related to this limitation, our estimates of calcium-dependent parameters, for example the ciliary reversal threshold, depend on an estimate of the effective ciliary volume. Compared to our estimate, based on electron microscopy measurements, this effective volume may be reduced by crowding, or increased by fast buffering. Changing this parameter results in proportional changes in concentration parameters. However, the fact that the fitted ciliary reversal threshold is close to the threshold measured on permeabilized *Paramecium* suggests that our estimate (also used in [25]) was reasonable.

Another limitation is we did not measure ciliary beating directly, but rather its effect on the fluid. This was motivated by the fact that we were interested primarily in the movement induced by ciliary beating, as well as by technical reasons. The constraints of measuring fluid motion while the cell is impaled with electrodes made it difficult to consistently achieve good spatial resolution over the entire recordings. In future work, high speed imaging of ciliary beating could be used to determine the spatial pattern of ciliary reversal with higher precision, although contractions of the cell may complicate the analysis.

Because of these technical limitations, our model of electromotor coupling was highly simplified, restricted to a phenomenological relation between calcium concentration and three kinetic parameters. It could also be that this relation is not instantaneous, involving more indirect pathways.

Finally, we considered only generic rather than biophysical models of sensory transduction, and we did not consider mechanical or hydrodynamic interactions with objects [72–74]. More generally, the behavioral repertoire of *Paramecium* includes other aspects that we did not attempt to model, such as the escape reaction [42,43], contractions [75], trichocyst discharge [76,77] and gravitaxis [78].

## How *Paramecium* turns

Since our model directly couples calcium concentration to kinetic parameters, it is not tied to any specific hypothesis about the ciliary beating pattern. However, turning is only possible if ciliary reversal is asynchronous, leading to a strongly asymmetrical ciliary beating pattern, otherwise the action potential would only trigger a back-and-forth movement in the direction of the main axis. We have shown that one possibility, compatible with the observed movement, is that cilia beat in a swirling pattern around the oral groove.

It is known that there is some structural and molecular heterogeneity between cilia, in particular between locomotor and oral cilia [9]. Whereas basal bodies are regularly placed on the dorsal side, they are spatially arranged on the ventral side with a characteristic pattern around

the oral groove (see Fig 8E). The beating frequency during helicoidal swimming is also spatially heterogeneous [79], and tail cilia are also known to be immobile [80]. Ciliary heterogeneity is likely to be more complex than a distinction between oral and locomotor cilia, because when *Paramecium* is cut in two pieces below the oral groove, both pieces can turn in a similar way [81]. Such heterogeneity is likely a general feature of motile microorganisms, some of which can exhibit complex gaits [82].

Machemer [59] and Párducz [60] described an asynchrony between anterior and posterior ciliary reversal, with anterior cilia returning to their initial beating direction before posterior cilia, which we confirmed with our PIV measurements. However, this is not sufficient to produce turning: if both anterior and posterior beating patterns are axisymmetric, any combination of them would still produce movement along the main axis. Jennings mentions that there is also a reversal asynchrony between the anterior left and anterior right cilia, such that all anterior cilia transiently beat towards the oral groove during the avoiding reaction [27]. More detailed investigation is necessary to clarify this question.

Physiologically, asynchronous ciliary reversal can be due to differential calcium sensitivity, that is, the calcium threshold for reversal might vary across cilia. This is the implicit assumption of our model. It could also be that there are differences in calcium entry or removal across cilia (e.g. a gradient of calcium channel expression, or calcium pumps, or calcium buffering molecules). Some studies suggest that cyclic nucleotides may also differentially regulate the reversal threshold [83,84].

## Previous models

We are aware of two previous attempts to model *Paramecium*'s action potential, neither of which was based on quantitative measurements. Hook and Hildebrand [85] used a calcium channel model with instantaneous transitions, an ohmic current-voltage relation (instead of GHK), an inactivation state accessible only from the closed state, and no voltage-dependent $K^+$ channel (only a model of $I_{K(Ca)}$, which is not the major $K^+$ current). Kunita et al. [86] used a Hodgkin-Huxley type model with voltage-dependent inactivation of calcium channels, which is not the main inactivation mode of this channel [31], even though the phenomenon exists on a slow timescale [87]. The model included two calcium channels (fast and slow), for which there is no electrophysiological support, and their relative activation was given as a function of time after stimulus start (i.e., it is not modeled). The calcium-dependent $K^+$ current was not included. Neither electrophysiological model was fitted to data, and neither was coupled to a kinematic model.

## Perspectives

The model could be improved by addressing the technical limitations listed above. In particular, it would be most enlightening to measure calcium concentration in the cilia at high temporal and spatial resolution, although it might require new technical developments. Further investigations should be carried out to understand in detail how *Paramecium* turns: to measure the spatial pattern of ciliary responses and to determine how this heterogeneity is achieved physiologically.

We have addressed only the avoiding reaction of *Paramecium*. The modeling effort could be completed by addressing other behavioral aspects, such as the escape reaction (increased ciliary beating speed upon hyperpolarization), which involves distinct hyperpolarization-activated channels [7]. *Paramecium* is sensitive to many sensory modalities, including temperature, various chemical substances, mechanical stimulation, light. Thus, the model should be completed by models of sensory transduction, as well as of mechanical interaction with

objects. This would allow us to use the model to investigate the physiological basis of behavior of *Paramecium* in complex environments.

Finally, this work opens the perspective of addressing complex autonomous behavior in ecological environments, including adaptation, learning and problem solving [7], with a systemic modeling approach.

## Materials and methods

### Paramecium culture and preparation

Cultures of *Paramecium tetraurelia* were obtained from Éric Meyer, Institut de Biologie, Ecole Normale Supérieure, Paris, France. For electrophysiological experiments (at Institut de la Vision), paramecia were co-cultured with *Klebsiella pneumoniae*, where each week 1 mL of culture was reinjected into 5 mL of Wheat Grass Powder (WGP) buffer supplemented with 1 μL of beta-sitosterol. Cultures were kept at room temperature (about 20°C). Cells were harvested in the early stationary growth phase, between 3 and 5 days after feeding them. To wash and concentrate cells for experiments, a droplet of culture (approximately 600 μL) was placed in a narrow neck volumetric flask before adding extracellular solution used for electrophysiology (see below). Due to negative gravitaxis [88], paramecia tend to accumulate at the top of the solution. Thus, after approximately 10 min, a concentrated population of cells were retrieved from the top of the flask and placed in a microcentrifuge tube for at least 3h for adaptation [50,89]. The tube was shaken before collecting cells to perform an experiment.

The culture method differed slightly for the behavioral measurements with freely swimming paramecia, because these cultures were done in another lab (Laboratoire Jean Perrin). Instead, before an experiment, bacteria were first grown in 5 mL of WGP for 24 h at 27°C, then paramecia were grown by adding 1 mL of *Paramecium* culture and 1 μL of beta-sitosterol to the bacterized WGP, for 48 h at 27°C in the dark. About 0.4 mL of cell suspension were then pipetted from the top of the culture tube into 4 mL of extracellular solution (see *Electrophysiology*), at least 20 minutes before an experiment.

### Swimming pools

Freely swimming paramecia were imaged at room temperature (~25°C) in square pools of side length 30 mm and depth 340 μm. These were obtained using micromilling and molding techniques. A Plexiglas mold, consisting of a square trench, is first milled with a square end mill of diameter 1 mm using a CNC micro-milling machine (Minitech, Machinary Corp., USA). Then a liquid mixture of Poly-DiMethyl Siloxane (PDMS, Sylgard 184, Dow Corning, USA) and its crosslinker (10:1 mass ratio) is poured onto the Plexiglas mold. It is immediately placed in a vacuum chamber for at least 1 h to remove any air bubbles. Crosslinking of the mixture is then obtained by placing the whole in an oven at 65°C for at least 4 h. Finally, the resulting transparent elastomer pool is gently peeled off the mold and put on a microscope glass slide. Prior to any experiment, the pool is exposed to an oxygen plasma for about 1 min to render the PDMS surface hydrophilic and prevent the trapping of air bubbles.

### Behavioral measurements

For all behavioral experiments, about 500 μL of the cell suspension is pipetted into the pool with a concentration of 300–600 cells/mL. Trajectories are imaged at 50 Hz with a CMOS camera (Blackfly S BFS-U3-51S5M-C, Flir, USA, 2448x2048 pixels$^2$, 10 bits), acquired with its dedicated acquisition software (Spinview, Flir, USA). A high magnification variable zoom lens (MVL12X12Z, Thorlabs) is used and yields a pixel size of 3.81 μm. The pool is uniformly

illuminated with a dark field configuration, by placing ~10 cm beneath the pool a square LED panel (EFFI-SBL, Effilux, France), on top of which a fully opaque mask is positioned, partially covering the LED panel (typically ¼ of its surface). The LED panel produces a red light (wavelength λ = 625 nm) to minimize phototaxis [90,91]. Movies of the swimming paramecia are 200 s long (see S7 Movie).

To limit hard drive space, images are stored without their background with lossless compression (TIFF format). The background image is computed by taking for each of its pixels the minimum pixel intensity over the first 100 frames. It is then subtracted from each frame, and pixels with an intensity value below a threshold (automatically computed with the "triangle method", see e.g. [92]) are set to 0.

Trajectories are extracted with the open source tracking software FastTrack [93], and manually inspected for corrections. Briefly, the software fits an ellipse to the cell's shape, and disambiguates front and rear based on the asymmetry of the pixel histogram along the main axis. Trajectories shorter than 1 s and sequences where the cell is immobile are discarded. Errors in front/rear identification are automatically corrected as follows: when the cell turns by more than 20˚ over two successive frames, it is considered an error and the angle is flipped.

Trajectories with circling motions are also discarded. To this end, we calculate the proportion of the trajectory where the cell turns clockwise versus anti-clockwise (for trajectories longer than 4 s). These proportions should be balanced (0.5) for helicoidal trajectories. The trajectory is eliminated if these proportions differ by more than 0.1 from the expectation, with manual confirmation. In total, there were n = 554 selected trajectories.

## Behavioral analysis

**Analysis of helicoidal trajectories.** We manually selected 20 trajectories presenting clear helicoidal motion in the focal plane from 2 experiments, totaling 121 s. In each helicoidal trajectory, cell orientation $\gamma(t)$ varies periodically with period $T$. We fitted $\gamma(t)$ to a sinusoidal signal. We found $T = 1.02 \pm 0.27$ $s$ (mean ± s.d.), corresponding to a spinning speed $\omega = \frac{1}{T} = 1.03 \pm 0.2$ cycle/s or about $2\pi$/s in radians. The amplitude was $\theta = 13 \pm 6.4$˚, the angle relative to the spiral axis.

**Analysis of avoiding reactions.** An avoiding reaction is defined as a portion of trajectory during which the cell swims backward. This backward swimming is detected when the instantaneous motion vector **m** and the orientation vector **o** (posterior to anterior) point to opposite directions, i.e., $\boldsymbol{m} \cdot \boldsymbol{o} < 0$. Avoiding reactions consisting of a single pair of frames were discarded. The mean frequency of spontaneous avoiding reactions was calculated as the number of avoiding reactions across all trajectories, divided by the total duration, yielding 0.18 Hz.

Only reorientation events involved in planar avoiding reactions were selected, based on measurements of the eccentricity of the ellipse that best fits the shape of paramecia. Whenever this eccentricity went below 0.8, the event was discarded. The total reorientation angle was obtained by summing all successive instantaneous reorientation angles during the entire avoiding reaction.

In Fig 7G, the 2D probability density of reorientation angle and backward duration was calculated with Gaussian kernel density estimation.

## Deciliation

Deciliated cells were obtained by adding 96% ethanol to a tube containing the previously washed and adapted cells in the extracellular solution up to a final concentration of 5% (v/v) [52]. Then the tube was shaken for 2 min and left to rest for 1 min. Deciliated cells were collected from the lower half of the tube since they no longer accumulate at the top of the

solution. Cilia start to grow back after approximately 30 min. Thus, as described in [52], in some experiments we blocked cilia regrowth by adding 10 mM of colchicine to the extracellular solution.

## Electrophysiology

The extracellular solution used in all experiments contains 1 mM $CaCl_2$, 4 mM KCl and 1 mM Tris-HCl buffer with pH 7.2, except for Fig 5B, where there was no KCl (blue curve). Microelectrodes of $\sim$ 50 M$\Omega$ resistance were pulled using a micropipette puller (P-1000, Sutter Instrument) from standard wall borosilicate capillary glass with filament (o.d. 1 mm, i.d. 0.5 mm, Harvard Apparatus). They were filled with a 1 M KCl solution using a MicroFil non-metallic syringe needle (MF 34G-5, World Precision Instruments); a few recordings were done with 3 M KCl (no particular change was noticed).

We used an upright microscope (LNScope, Luigs & Newmann) with two objectives, a 20× air objective (SLMPLN Plan Achromat, Olympus) to locate cells, and a 40× water immersion objective (LUMPLFLN, Olympus) with DIC contrast enhancement for electrophysiology and imaging.

Paramecia were immobilized using the device described in [22]. Briefly, paramecia are immobilized against a transparent filter (Whatman Cyclopore polycarbonate membranes; diameter 25 mm, pore diameter 12 μm) thanks to a peristaltic pump (Gilson Minipulse 3) that circulates the fluid from below the filter to above the device. Two microelectrodes are then lowered into the cell, and the pump is stopped. The cell is then held in place by the electrodes.

Electrophysiology recordings were performed using an amplifier with capacitance neutralization (Axoclamp 2B and Axoclamp 900 A, Molecular Device) and an analog–digital acquisition board operating at a sampling frequency of 40 kHz (USB-6343, National Instruments). Custom Python programs (https://github.com/romainbrette/clampy) were used to control the acquisition board.

Membrane potential was recorded with the reading electrode while 100 ms current pulses of various amplitudes were injected through the second electrode, with at least 1 s between successive trials.

## Particle image velocimetry

To measure the flows induced by cilia beating, the bath was seeded with 1 μm silica or polystyrene particles (~0.2 mM) after paramecia were immobilized and the pump was stopped. Because of sedimentation, particle density was typically higher at the beginning of the experiment. Images were recorded at 30 Hz with a high-sensitivity CCD camera (Lumenera Infinity 3-6UR) over a 1392×1392 pixels region of interest surrounding the cell (8 bits depth, pixel width 0.178 μm). Frames were synchronized with electrophysiology recordings using a digital trigger.

Frames were preprocessed by removing the background (average image) and band-pass filtering (subtraction of two Gaussian filters with standard deviation 1 μm and 1.3 μm). Consecutive frames were then analyzed with particle image velocimetry (PIV) using the OpenPIV Python package (https://github.com/OpenPIV/openpiv-python.git), which calculates the velocity field using image cross-correlation. We used 50 μm windows with 2/3 overlap.

In each frame, we calculated the mean angle of the velocity vector over the entire field, using circular mean (argument of the mean complex unit vector; occasional missed frames were discarded). We then subtracted the angle of the anteroposterior axis. The position of anterior and posterior ends was measured manually. As the two ends can be visually

ambiguous, they were automatically corrected (by swapping) when the flow measured before stimulation was directed towards the anterior end (indicating backward swimming).

In Fig 8A, for each cell we averaged the mean angle over all currents and over the 300 ms before stimulus (blue) or over the second half of the stimulus (red), for positive currents (<5 nA).

We also calculated the mean angle in the anterior and posterior regions as indicated in Fig 8G. Each region is a half-plane orthogonal to the main axis, starting at one end. For this analysis, we selected n = 9 cells with high quality video recordings and clear cell positioning, indicated by an absence of missed frames and a pre-stimulus flow deviating by less than 45˚ from the main cell axis. The average was restricted to responses to large pulses (1 to 5 nA), because those were recorded before the small pulses and therefore had a higher density of particles (due to sedimentation).

## Electrophysiological modeling

In this section, we describe the biophysical models. The parameter values are obtained by fitting the models to the data (section *Model fitting*).

**Electrode model.**   All recordings were done with two electrodes, an injecting electrode and a reading electrode. Because of the capacitance and resistance of the injecting electrode, the current injected in the cell is a low-pass filtered version of the command current [94]. To estimate this current, we model the injecting electrode as a simple RC circuit and estimate its parameters $R_e$ and $\tau_e$ from responses to small pulses, assuming passive cell responses:

$$C\frac{dV}{dt} = -g_L(V - V_0) + I_e \tag{1}$$

$$\tau_e \frac{dV_2}{dt} = V - V_2 + R_e I + \Delta V \tag{2}$$

$$I_e = \frac{V_2 - V - \Delta V}{R_e} \tag{3}$$

where $V$ is the membrane potential, assumed identical to the potential of the reading electrode, $V_2$ is the potential of the injecting electrode, $I_e$ is the current injected in the cell, and $\Delta V$ accounts for a difference in tip potentials. The membrane equation (first equation) is a rough linear model of the cell, but only parameters $R_e$ and $\tau_e$ are used subsequently, to estimate $I_e$ from $I$ according to the last two equations ($\Delta V$ has no impact on $I_e$ and therefore can then be discarded). In the 29 ciliated cells analyzed for passive properties, we found $R_e$ = 121 ± 8 MΩ and $\tau_e$ = 1 ± 0.9 ms.

**Ionic currents.**   *Paramecium* electrophysiology is reviewed in [18] and updated in [7,95]. *Paramecium* in an isopotential cell [23,24]. Thus, we consider a single membrane equation:

$$C\frac{dV}{dt} = I_L + I_{Kir} + I_{Kd} + I_{Ca} + I_{K(Ca)} + I$$

where $C$ is membrane capacitance, $I_L = g_L(E_L - V)$ is the leak current, $I_{Kd}$ is the delayed rectifier K$^+$ current responsible for repolarization, $I_{Ca}$ is the ciliary voltage-dependent Ca$^{2+}$ current, $I_{K(Ca)}$ is the calcium-activated K$^+$ current, $I_{Kir}$ is the inward rectifier K$^+$ current and $I$ is a stimulating current. We did not include a few other electrophysiologically identified currents that are less relevant for this study, namely: Na$^+$ [96,97] and Mg$^{2+}$ [98,99] currents (since our extracellular solution does not contain these two ion species), and hyperpolarization-activated calcium currents responsible for the escape reaction [37–39], which we did not model.

The inward rectifier current $I_{Kir}$ is a $K^+$ current activated by hyperpolarization, most strongly below $E_K$ [40]. It is modeled as follows:

$$I_{Kir} = g_{Kir} n_{Kir}^p (E_K - V) \tag{4}$$

$$\tau_{Kir} \frac{dn_{Kir}}{dt} = \frac{1}{1 + \exp\left(\frac{V - V_{Kir}}{k_{Kir}}\right)} - n_{Kir} \tag{5}$$

where $p = 1$ or $2$ ($p = 2$ in the final version). We made this simple modeling choice because this current was only used as a way to infer the reversal potential $E_K$. In particular, we did not include inactivation [41]. We also tested a version of the model where the linear driving force $(E_K - V)$ is replaced by the Goldman-Hodgkin-Katz expression [54], but it made no significant difference in fitting results.

For the delayed rectifier current $I_{Kd}$, we tested several models of the type:

$$I_{Kd} = g_{Kd} n^p (E_K - V) \tag{6}$$

$$\tau_{Kd} \frac{dn}{dt} = n_\infty(V) - n \tag{7}$$

We tested two classes of models. The Hodgkin-Huxley model is:

$$\alpha_n(V) = \frac{a_{Kd}}{\mathrm{exprel}\left(\frac{V_{Kd} - V}{k_{Kd}^a}\right)} \tag{8}$$

$$\beta_n(V) = b_{IK} \exp\left(\frac{V_{Kd} - V}{k_{Kd}^b}\right) \tag{9}$$

$$n_\infty(V) = \frac{\alpha_n(V)}{\alpha_n(V) + \beta_n(V)} \tag{10}$$

$$\tau_{Kd}(V) = \tau_{Kd}^{min} + \frac{1}{\alpha_n(V) + \beta_n(V)} \tag{11}$$

where

$$\mathrm{exprel}(x) = (e^x - 1)/x$$

For numerical stability (near $x = 0$), we use this special function in the code rather than the explicit expression.

The Boltzmann model is:

$$n_\infty(V) = \frac{1}{1 + \exp\left(\frac{V_{Kd} - V}{k_{Kd}}\right)} \tag{12}$$

$$\tau_{Kd}(V) = a_{Kd} + \frac{b_{Kd}}{\cosh\left(\frac{V - V_\tau^{Kd}}{k_\tau^{Kd}}\right)} \tag{13}$$

The voltage-gated calcium current $I_{Ca}$ is a calcium-inactivated current located in the cilia [18,31]. The corresponding channels have been genetically identified; they are similar to the Cav1 mammalian family (L-type), with a putative calmodulin binding site [16]. We model it similarly to [53,100]:

$$I_{Ca} = g_{Ca} m^2 h f_{GHK}(V) \tag{14}$$

where $m$ is the activation gating variable, $h$ is the inactivation gating variable, and $f_{GHK}(V)$ is the normalized current-voltage relation of the open channel, given by the Goldman-Hodgkin-Katz equation [54]. In the Hodgkin-Huxley model, this relation is linear. However, with very different intracellular and extracellular calcium concentrations, a better model is the Goldman-Hodgkin-Katz equation. Resting intracellular concentration is about 50–200 nM [55,56], and rises to an estimated 20 μM during an action potential [25]. In contrast, extracellular concentration is 1mM in our experiments. Thus, we neglect intracellular concentration, which yields:

$$f_{GHK}(V) = \frac{1}{\mathrm{exprel}(2FV/RT)}$$

where $F$ is the Faraday constant, $R$ is the gas constant, and $T = 293$ K is temperature (20°C). Here, extracellular concentration has been lumped into $g_{Ca}$, which is now homogeneous to a current, while $f_{GHK}(V)$ is unitless and has no free parameter.

The activation gating variable is governed by:

$$\tau_m \frac{dm}{dt} = \frac{1}{1 + \exp\left(\frac{V_{Ca} - V}{k_{Ca}}\right)} - m \tag{15}$$

while the inactivation gating variable is a Hill function of intraciliary calcium concentration $[Ca^{2+}]$:

$$h([Ca^{2+}]) = \frac{1}{1 + \left(\frac{[Ca^{2+}]}{K_{Ca}}\right)^{n_{Ca}}} \tag{16}$$

This is similar to the model of [53], except that the number of sites $n_{Ca}$ is allowed to be greater than 1, because we found that this was necessary to fit our data.

A calcium-activated K$^+$ current has been identified by comparison with Pawn mutants lacking functional voltage-activated calcium currents [36]. The current is largely reduced by EGTA. Genomic analysis indicates the presence of both BK and SK channels, with SK channels immunochemically identified in the cilia [12,34]. We model the current as follows, with activation as a Hill function of $[Ca^{2+}]$:

$$I_{K(Ca)} = g_{K(Ca)} m([Ca^{2+}])(E_K - V) \tag{17}$$

$$m([Ca^{2+}]) = \frac{1}{1 + \left(\frac{K_{K(Ca)}}{[Ca^{2+}]}\right)^{n_{K(Ca)}}} \tag{18}$$

**Calcium dynamics.** Resting intracellular calcium concentration $[Ca^{2+}]_0$ has been estimated between 50 and 200 nM [55,56]. We chose $[Ca^{2+}]_0 = 100$ nM. Calcium enters the cilia when calcium channels open. The concentration increase is spatially uniform along the cilium

[101]. It then decreases by three mechanisms: buffering, pumps, and diffusion. Buffering can occur with a variety of calcium-binding proteins, an important one being centrin, located in the infraciliary lattice, at the base of cilia [102,103]. Plasma membrane calcium pumps (PMCA) have been identified in the basal membrane with low affinity, around $10^{-7}$ M [104], and also in the cilia [33,34]. Suppressing the ciliary PCMAs by RNA interference prolongs backward swimming, which means that they are involved in the removal of calcium after an action potential. In principle, calcium can also diffuse to the basal cytosol. However, this has not been observed [105]. This might be because of cilia volume compared to the cell, or because calcium is buffered at the base of cilia. Both phenomena can be modeled by diffusion to the cilium base, with fixed resting concentration at the boundary.

We lump these diverse mechanisms into two simple processes: a linear process, with rate proportional to $([Ca^{2+}]-[Ca^{2+}]_0)$, modelling diffusion and low-affinity buffers, and a high-affinity process operating near rest, with rate given by a Hill function of $[Ca^{2+}]$, modelling PMCAs or high affinity buffers. This results in the following equation:

$$\frac{d[Ca^{2+}]}{dt} = \frac{I_{Ca}}{2Fv_{\text{cilia}}} - \lambda\left([Ca^{2+}] - [Ca^{2+}]_0\right) - \frac{J}{1 + \frac{[Ca^{2+}]_0}{[Ca^{2+}]}} \tag{19}$$

where $v$ is the volume of cilia and F is the Faraday constant. It can be seen that the role of the high-affinity process in this model is to counteract the calcium flow at rest, namely $J = I_{rest}/Fv_{\text{cilia}}$, while the low-affinity process independently tunes the rate of calcium removal after an action potential.

There are 3000–4000 cilia in *P. tetraurelia* [8–10], with the upper estimates likely including oral cilia. Each cilium is 10–12 μm long [23,106]. Each cilium is 270 nm wide but the fiber bundle is 200 nm wide [9]. This yields a total volume between 950 and 2750 μm³. We used the previous estimate $v_{\text{cilia}} = 1700$ μm³ from [25], which is compatible with these bounds, but the uncertainty is large. In addition, the effective volume might be smaller because of crowding, or larger because of fast buffering. In practice, an error in the estimation of ciliary volume will translate into an equivalent change in all calcium binding constants (as well as $\lambda$ and $J$). Binding constants and volume cannot be determined independently, because the (inverse) volume effectively acts as a unit for those constants.

For stability, the numerical implementation of calcium-dependent equations used equivalent versions written as a function of

$$p_{Ca} \equiv \log \frac{[Ca^{2+}]}{[Ca^{2+}]_0}$$

where $[Ca]_0 = 0.1$ μM is the resting concentration. For example, inactivation is rewritten as:

$$h([Ca^{2+}]) = \frac{1}{1 + \exp(n_{Ca}(p_{Ca} - p_{K_{Ca}}))}$$

This equivalent change of variables avoids numerical issues when $[Ca^{2+}]$ approaches 0. The calcium dynamics equation rewrites as follows:

$$\frac{dp_{Ca}}{dt} = \frac{I_{Ca}}{2F[Ca^{2+}]_0 v_{\text{cilia}}} e^{-pCa} + \lambda(e^{-pCa} - 1) - \frac{J}{1 + e^{pCa}}$$

**Electromotor coupling.** Cilia reorient when intraciliary calcium concentration reaches about 1 μM [30]. We model the ciliary angle as a Hill function of $[Ca^{2+}]$:

$$\alpha = \alpha_0 + \frac{\Delta\alpha}{1 + \left(\frac{K_{motor}}{[Ca^{2+}]}\right)^{n_{motor}}} \tag{20}$$

where angles are relative to the anteroposterior direction ($\alpha = 0$ when cilia beat to the rear), and $K_{motor}$ is the reversal threshold.

Velocity is modeled as an affine transformation of a Hill function with coefficient $n = 2$, changing sign at $K_{motor}$:

$$v = -v_{max} + \frac{2v_{max}}{1 + \left(\frac{[Ca^{2+}]}{K_{motor}}\right)^2} \tag{21}$$

where $v_{max} = 500$ μm/s is maximum velocity (both backward and forward), according to our measurements.

The angle $\theta$ of the rotation axis and the spinning speed $\omega$ are modeled as bell functions peaking at $K_{motor}$:

$$\theta = \theta_{min} + 2\frac{\theta_{max} - \theta_{min}}{\left(\frac{K_{motor}}{[Ca^{2+}]}\right)^2 + \left(\frac{[Ca^{2+}]}{K_{motor}}\right)^2} \tag{22}$$

$$\omega = \omega_{min} + 2\frac{\omega_{max} - \omega_{min}}{\left(\frac{K_{motor}}{[Ca^{2+}]}\right)^2 + \left(\frac{[Ca^{2+}]}{K_{motor}}\right)^2} \tag{23}$$

We set $\theta_{min} = 13°$ based on our measurements, and $\theta_{max} = 90°$, to allow for planar rotations as illustrated in Fig 8D. We set $\omega_{min} = 2\pi/s$ (1 cycle/s) based on our measurements and $\omega_{min} = 8\pi/s$ (2 cycles/s), based on the doubling found in the spherical model (Fig 8F). Thus, no extra free parameter is introduced.

**Sensory transduction.** *Instantaneous transduction*. In Fig 10A and 10B where a well-delimited object (half-plane or disc) acts as a stimulus, we first calculate the proportion of the cell surface in contact with the object. To this end, cell shape is determined by the formula proposed by Zhang et al. [107]:

$$y(x) = \frac{b}{2}\left(\sqrt{1 - 4\frac{x^2}{a^2}} - \beta\sin\left(\frac{2\pi x}{a}\right)\right)$$

where $x$ is the position along the major axis and $y$ the position along the minor axis, $a = 120$ μm is cell length, $b = 35$ μm is cell width and $\beta = 0.15$ is an asymmetry factor. We then simply calculate the intersection of cell and object shapes (as pixel images). The stimulus current is then $I = I_0 p$, where $I_0$ is maximum current and $p$ is the fraction of the cell image within the object.

*Delayed transduction*. In Fig 10B and 10A, we simply consider that the transduction current activates and deactivates with a time constant $\tau_I$:

$$\tau_I \frac{dI}{dt} = I_0 p - I \tag{24}$$

This simple model corresponds to channels with finite opening and closing rates (namely, opening rate $\alpha = \tau_I s$ and closing rate $\beta = \tau_I(1 - s)$, where $s$ is proportional to the stimulus $I_0 p$) but does not take into account the spatial recruitment of channels.

*Adaptation*. In Fig 11B, 11C and 11D, we consider that there are two pathways with opposite polarity, a fast pathway and a slower pathway:

$$\tau_{\text{fast}} dI_{\text{fast}} = s - I_{\text{fast}} \tag{25}$$

$$\tau_{\text{slow}} dI_{\text{slow}} = s - I_{\text{slow}} \tag{26}$$

$$I = I_{\text{slow}} - I_{\text{fast}} \tag{27}$$

where $\tau_{\text{fast}} = 40$ ms and $\tau_{\text{slow}} = 200$ ms. Thus, for a constant stimulus $s$, the stationary current is 0. In Fig 11B (disc stimulus), the stimulus is $s = I_0 p$. In Fig 11C and 11D, where the environment is spatially continuous, the stimulus is simply the value at the center of the cell.

## Model optimization

Model parameters are estimated with the model fitting toolbox of the Brian simulator [45,46] (https://github.com/brian-team/brian2modelfitting). Briefly, the software performs least square optimization using a combination of global optimization algorithms (we used differential evolution) and gradient descent, where the gradient is calculated symbolically from the model equations. Optimization with multiple objectives is done by adding the errors associated to the different objectives. Compiled code is automatically produced by code generation. Each fitting procedure took up to a few hours, and fitting scripts were run in parallel on different cells, using a small cluster of 3 PC with 8 cores each.

After model fitting, cells were discarded if passive properties were abnormal, indicating a bad recording (C > 500 pF or R<30 MΩ or R>500 MΩ), or if $E_K > E_L$ (which is biophysically impossible), indicating a fitting problem.

*Electrode fitting*. First, for each cell we estimated electrode resistance $R_e$ and time constant $\tau_e$ from responses to small 100 ms pulses, both hyperpolarizing and depolarizing (|I|<0.5 nA), using Eqs (1)–(3). The error criterion was the sum of quadratic errors on both electrode potentials, measured from 100 ms before to 100 ms after the pulse. The estimated parameters were then used in subsequent fits.

*Hyperpolarized fits*. For Fig 5, we fitted the models described above with $I_L$ and $I_{\text{Kir}}$, (Eqs (4) and (5)), with least square minimization of the error on the reading electrode potential, taken from pulse start to 50 ms after the pulse. The stimuli were 100 ms pulses with amplitude between -4 and 0 nA in 300 pA increments.

We fitted the model with $p = 1$ and with $p = 2$ for $I_{Kir}$. Using two gates ($n^2$) gave better fits than using one (n = 40; median error 2.3 vs. 2.6 mV; p = $8.10^{-5}$, one-tailed Wilcoxon test). We also fitted the $n^2$ model with the ohmic driving force replaced by a Goldman-Hodgkin-Katz model, but it did not yield any significant improvement (p = 0.27, two-tailed Wilcoxon test). Selection criteria (see above) were passed by n = 28 cells. Statistics of fitted parameters are shown in Table 1.

**Table 1. Statistics of fitted parameters (n = 28) for hyperpolarized responses (model with two gates).**

| | Mean | Median | s.d. | s.e.m. |
|---|---|---|---|---|
| **C** (pF) | 288.87 | 278.82 | 75.32 | 18.83 |
| **$E_K$** (mV) | -48.27 | -47.92 | 9.54 | 2.38 |
| **$E_L$** (mV) | -20.32 | -19.03 | 11.32 | 2.83 |
| **$V_{Kir}$** (mV) | -130.27 | -120.75 | 35.36 | 8.84 |
| **$g_L$** (nS) | 9.41 | 7.11 | 6.93 | 1.73 |
| **$g_{Kir}$** (nS) | 1375.38 | 873.36 | 1206.00 | 301.50 |
| **$k_{Kir}$** (mV) | 31.53 | 30.23 | 8.89 | 2.22 |
| **$\tau_{Kir}$** (ms) | 15.44 | 15.10 | 3.69 | 0.92 |
| **1/R** (nS) | 1.21 | 1.14 | 0.72 | 0.18 |
| **$V_0$** (mV) | -24.56 | -22.51 | 10.64 | 2.66 |

*Deciliated fits.* For Fig 6, we fitted the models described above with $I_L$ and $I_{Kd}$, with least square minimization of the error on the reading electrode potential, taken from pulse start to 50 ms after the pulse. The stimuli were 100 ms pulses with amplitude between 0 and 4 nA in 300 pA increments. Parameters $E_K$ and $C$ were taken from the previous fit to hyperpolarized responses. To make sure the short onset is well captured, the time interval is split in two windows: the first 30 ms and the rest of the response, and each window is equally weighted (meaning that a data point in the first window contributes more than a data point in the second window).

We tested the Hodgkin-Huxley (HH) type model (Eqs (6)–(11)) and the Boltzmann model (Eqs (6), (7), (12) and (13)). The two models performed similarly (n = 21; median 1.78 mV vs. 1.69 mV; p = 0.59, two-tailed Wilcoxon test). The median number of gates was 1.13 in the HH model (1.1–1.8, 25–75% interval) and 1.7 (1.3–2.7) in the Boltzmann model. In both models, the minimal time constant was very small (median 0.1 vs. 0.6 ms). Therefore, we chose a model with 2 gates ($n^2$) and a nearly null minimal time constant (100 μs for numerical reasons). A Boltzmann $n^2$ model gave similar results to a HH $n^2$ model (p = 0.96, two-tailed Wilcoxon test) and had fewer parameters, while performing similarly to the unconstrained model (p = 0.79, two-tailed Wilcoxon test). Therefore, we chose the Boltzmann $n^2$ model. Selection criteria (see above) were passed by n = 16 cells. Statistics of fitted parameters are shown in Table 2.

**Table 2. Statistics of fitted parameters (n = 16) for deciliated cells (Boltzmann model with two gates).**

| | Mean | Median | s.d. | s.e.m. |
|---|---|---|---|---|
| **C** (pF) | 158.59 | 145.61 | 48.76 | 12.19 |
| **$E_K$** (mV) | -53.83 | -56.38 | 12.35 | 3.09 |
| **$E_L$** (mV) | -19.68 | -18.33 | 6.48 | 1.62 |
| **$V_{Kd}$** (mV) | 29.77 | 21.31 | 22.86 | 5.71 |
| **$V_{\tau_{Kd}}$** (mV) | 26.32 | 23.09 | 48.37 | 12.09 |
| **$b_{Kd}$** (ms) | 9.21 | 4.40 | 12.69 | 3.17 |
| **$g_L$** (nS) | 10.62 | 9.72 | 5.73 | 1.43 |
| **$g_{Kd}$** (nS) | 1130.25 | 100.11 | 2656.07 | 664.02 |
| **$k_{Kd}$** (mV) | 8.96 | 6.61 | 6.29 | 1.57 |
| **$k_{\tau_{Kd}}$** (mV) | 13.54 | 11.74 | 8.08 | 2.02 |

Fitting results motivated us to further simplify the model by enforcing $V_{Kd} = V_\tau$ and $k_\tau = 2k_{Kd}$. This corresponds to a simple biophysical model where opening and closing rates are of the form $e^{\pm V/k}$. This simplification slightly increases the fit error (1.82 vs. 1.8 mV; p = 0.009, two-tailed Wilcoxon test), but has the advantage of reducing the parameter set to a single kinetic parameter, the maximum time constant (median 3.2 ms).

*Ciliated fits.* For Fig 7, we fitted the complete model to electrophysiological and ciliary responses to two sets of 100 ms pulses, a set of pulses between 0 and 5 nA in 300 pA increments, and a set of pulses between -100 and 500 pA in 25 pA increments. The complete model consisted of $I_L$, the simplified $n^2$ Boltzmann model of $I_{Kd}$ with $a_{Kd} = 0.1$ ms, $V_{K_d} = V_\tau^{Kd}$ and $k_\tau^{Kd} = 2k_{Kd}$ (Eqs (6), (7), (12) and (13)), $I_{Ca}$ (Eqs (14)–(16)), $I_{K(Ca)}$ (Eqs (17) and (18)), calcium dynamics (Eq (19)) and electromotor coupling (Eq (20)).

To deal with possible shifts in tip potential between the two sets, we aligned all traces to a resting potential of -22 mV (the median resting potential), and $E_K$ was fixed at -48 mV (the median estimated $E_K$).

The optimization error combined an error on the reading electrode potential, an error on the ciliary angle (mean angle of the PIV analysis), and an error on resting calcium concentration. For the voltage error, the response was divided in two equally weighted intervals: from pulse start to pulse end, and from pulse end to 500 ms after the end (to capture the post-stimulus hyperpolarization). The angle error was defined as the quadratic error on the corresponding unit vectors, which is equivalent to $E_\alpha = \left(\cos(\alpha) - \cos(\widehat{\alpha})\right)^2 + \left(\sin(\alpha) - \sin(\widehat{\alpha})\right)^2$, and applied on the interval from 100 ms before the pulse to 500 ms after it. Finally, we ensured that the resting $[Ca^{2+}]$ was 0.1 μM by inserting an error on $[Ca^{2+}]$ on the interval from 100 ms before the pulse to the start of the pulse ($E_{Ca} = ([Ca^{2+}] - 0.1\ \mu M)^2$). This effectively ensures $J = I_{rest}/Fv_{cilia}$.

Selection criteria (see above) were passed by n = 18 cells. Statistics of fitted parameters are shown in Table 3.

## Statistics

Fitting results obtained with different models were compared using the Wilcoxon test. Statistics are given as mean ± standard deviation. In box plots, the box shows the first and third quartile with the median value inside, and the whiskers are the minimum and maximum values excluding outliers, which are shown as diamonds and defined as those at a distance exceeding 1.5 times the interquartile range from the box.

## Hydrodynamic model

In Fig 8F, we calculated the motion vectors from patterns of ciliary beating on a sphere of 60 μm radius. The velocity vector $U$ and the rotation vector $\Omega$ are given by:

$$\begin{pmatrix} U \\ \Omega \end{pmatrix} = \begin{pmatrix} M & N \\ N^T & O \end{pmatrix} \begin{pmatrix} F \\ L \end{pmatrix}$$

where **F** is the external force and **L** is the external torque [58]. The matrix is called the *mobility* matrix. For a sphere of radius $r$, the mobility matrix is diagonal:

$$U = (6\pi\eta r)^{-1} F$$

$$\Omega = (8\pi\eta r^3)^{-1} L$$

**Table 3. Statistics of fitted parameters (n = 16) for depolarized ciliated cells.**

| | Mean | Median | s.d. | s.e.m. |
|---|---|---|---|---|
| **C** (pF) | 303.06 | 289.10 | 67.81 | 15.98 |
| **E$_L$** (mV) | -23.07 | -23.30 | 1.18 | 0.28 |
| **J** (1/s) | 946.72 | 866.02 | 422.76 | 99.65 |
| **V$_{Ca}$** (mV) | 0.47 | -1.33 | 6.03 | 1.42 |
| **V$_{Kd}$** (mV) | 9.55 | 4.19 | 12.48 | 2.94 |
| **$\lambda$** (1/s) | 19.43 | 7.79 | 32.31 | 7.62 |
| **b$_{Kd}$** (ms) | 4.77 | 4.50 | 2.58 | 0.61 |
| **g$_{Ca}$** (nA) | 434.81 | 226.63 | 413.79 | 97.53 |
| **g$_{K(Ca)}$** (nS) | 3919.39 | 216.51 | 6499.27 | 1531.89 |
| **g$_L$** (nS) | 10.38 | 9.52 | 4.78 | 1.13 |
| **g$_{Kd}$** (nS) | 821.80 | 114.76 | 1469.93 | 346.47 |
| **k$_{Ca}$** (mV) | 4.34 | 4.35 | 0.97 | 0.23 |
| **k$_{Kd}$** (mV) | 5.74 | 4.89 | 2.97 | 0.70 |
| **n$_{Ca}$** | 4.41 | 4.25 | 1.66 | 0.39 |
| **n$_{K(Ca)}$** | 3.53 | 2.94 | 2.35 | 0.55 |
| **n$_{motor}$** | 7.37 | 7.02 | 5.70 | 1.34 |
| **pK$_{Ca}$** | 3.60 | 3.61 | 0.40 | 0.10 |
| **pK$_{K(Ca)}$** | 6.37 | 6.79 | 1.50 | 0.35 |
| **pK$_{motor}$** | 2.64 | 3.14 | 1.87 | 0.44 |
| **$\tau_m$** (ms) | 0.91 | 0.94 | 0.38 | 0.09 |

We consider that each patch of membrane is subjected to a force from the fluid, in the direction opposite to the ciliary beating direction, and we calculate the total force and torque for different ciliary beating patterns.

We use spherical coordinates $(\theta, \phi)$ (see Fig 12), where $\theta = 0$ corresponds to the North pole, considered as the anterior end, and $\phi = 0$ is the meridian corresponding to the oral groove. Thus, a surface element is:

$$dS = r^2 \sin \theta \, d\theta d\phi$$

The local force is tangent to the sphere and oriented with an angle $\alpha$, where $\alpha = 0$ is the direction of the meridian, pointing South. Cartesian coordinates are chosen so that the $x$ axis is dorso-ventral, the $y$ axis is oriented left to right, and the $z$ axis is posterior to anterior (i.e., South to North pole).

With these conventions, the local force expressed in Cartesian coordinates is:

$$\boldsymbol{F} = A \begin{bmatrix} \cos \theta \cos \theta \cos \phi - \sin \alpha \sin \phi \\ \cos \theta \cos \alpha \sin \phi + \sin \alpha \cos \phi \\ -\sin \theta \cos \alpha(\theta, \phi) \end{bmatrix}$$

where $A$ is the amplitude of the force per unit area and $\alpha$ is its angle (to obtain this result, start from the North pole, rotate along the $y$ axis by $\theta$, then rotate along the $z$ axis by $\alpha$). Therefore,

the total force is:

$$\boldsymbol{F}_{tot} = r^2 \int_{-\pi}^{\pi}\int_{0}^{\pi} \boldsymbol{F}(\theta, \phi) \sin\theta \, d\theta d\phi$$

The local torque is $\boldsymbol{\tau} = \boldsymbol{r} \times \boldsymbol{F}$ where $\boldsymbol{r}$ is a radius. In Cartesian coordinates, we obtain:

$$\boldsymbol{\tau} = rA \begin{bmatrix} -\cos\theta\sin\alpha\cos\phi - \cos\alpha\sin\phi \\ -\cos\theta\sin\alpha\sin\phi + \cos\alpha\cos\phi \\ \sin\theta\sin\alpha \end{bmatrix}$$

$$\boldsymbol{\tau}_{tot} = r^2 \int_{-\pi}^{\pi}\int_{0}^{\pi} \boldsymbol{\tau}(\theta, \phi) \sin\theta \, d\theta d\phi$$

The reported velocity is $v = \|\boldsymbol{U}\|$, the angle of the rotation axis $\theta_{\text{rotation}}$ is calculated in the xz plane, and the spinning speed is $\omega = \frac{\|\boldsymbol{\Omega}\|}{2\pi}$ in cycle/s. Local force amplitude is identical in all ciliary beating patterns, and chosen so as to obtain a forward velocity of 500 μm/s.

Three different patterns are represented in Fig 8F. In the forward pattern, the local angle is uniform: $\alpha$ = -170°, corresponding to a beating direction of 10°, downward to the right. In the backward pattern, the local angle is $\alpha$ = -10°, corresponding to a beating direction of 170°, upward to the right (obtained by up/down symmetry). In both cases, the local force is axisymmetrical with respect to the anteroposterior axis. It follows that both $\boldsymbol{F}_{tot}$ and $\boldsymbol{\tau}_{tot}$ are aligned with the z axis (anteroposterior). This can be seen in the formulas above by integrating with respect to $\phi$, which yields 0 for the $x$ and $y$ coordinates.

In the turning pattern, the left anterior quarter ($\phi \in [-\pi, 0]$, $\theta \in \left[0, \frac{\pi}{2}\right]$) follows the forward pattern ($\alpha$ = -170°), while the right anterior quarter ($\phi \in [0, \pi]$, $\theta \in \left[0, \frac{\pi}{2}\right]$) follows the backward pattern ($\alpha$ = -10°) and the posterior half ($\theta \in [\pi/2, \pi]$) has local forces pointing to the left ($\alpha$ = -90°), meaning cilia beating to the right. The posterior half generates a rotating pattern around the main axis (by axisymmetry), without translational movement. Each anterior quarter generates forces and torques

$$\boldsymbol{F}_{tot} = \begin{bmatrix} 2\sin\alpha \\ -\cos\alpha \\ -\frac{\pi}{4}\cos\alpha \end{bmatrix}$$

$$\boldsymbol{\tau}_{tot} = r \begin{bmatrix} 2\cos\alpha \\ \sin\alpha \\ \frac{\pi}{4}\sin\alpha \end{bmatrix}$$

With $\alpha_L = \pi - \alpha_R$, we then find that the $y$ and $z$ components of the total torque vanish. Added to the torque generated by the posterior part, we obtain a torque vector in the $xz$ plane, which is the plane of the oral groove, separating the cell in left and right parts.

## Kinematics

We consider that the organism is an object moving by rigid motion. The organism is characterized by a position vector **x**, and an orientation matrix **R** defining the rotation of the

reference frame (frame of the organism), so that a point **y** on the reference frame is mapped to **Ry** in the observer frame. The reference frame is chosen as in the spherical model above, so that $z > 0$ points towards the anterior end while the $x$ axis the dorsoventral axis.

Translational velocity is assumed to be in the posterior-anterior direction only: $\boldsymbol{v} = [0, 0, v]$, so that

$$\dot{\boldsymbol{x}} = \boldsymbol{R}v$$

Let $\boldsymbol{\omega}$ be the rotation vector in the reference frame. We assume it is tilted from the main axis by an angle $\theta$ in the $xz$ plane:

$$\boldsymbol{\omega} = -\omega[\sin(\theta), 0, \cos(\theta)]$$

Thus, kinematics is determined by three variables $v$, $\omega$ and $\theta$, which are functions of calcium concentration, as detailed in *Electrophysiological modeling (subsection Electromotor coupling)*.

Over a time $dt$, the organism rotates by $\Omega(dt) = \boldsymbol{I} + [\boldsymbol{\omega}].\, dt$, where

$$[\boldsymbol{\omega}] = \omega \begin{bmatrix} 0 & -\cos(\theta) & 0 \\ \cos(\theta) & 0 & -\sin(\theta) \\ 0 & \sin(\theta) & 0 \end{bmatrix}$$

is the infinitesimal rotation matrix, such that $[\boldsymbol{\omega}]\mathbf{y} = \boldsymbol{\omega} \times \mathbf{y}$. Therefore, the orientation matrix changes as $\boldsymbol{R}(t + dt) = \boldsymbol{R}\Omega(dt)$, giving:

$$\dot{\boldsymbol{R}} = \boldsymbol{R}[\boldsymbol{\omega}]$$

For numerical reasons, we use quaternions instead of matrices [108], implemented with the Python packages *quaternion* and *pyquaternion*. Orientation is then represented by a unit quaternion $q$, and kinematic equations translate to:

$$\dot{\widehat{\boldsymbol{x}}} = q\widehat{\boldsymbol{v}}\bar{q}$$

where $\widehat{\boldsymbol{v}}$ is the pure quaternion with imaginary part **v**, and

$$\dot{q} = \frac{1}{2}q\widehat{\boldsymbol{\omega}}$$

*Confinement to a plane*. We constrain the organism to move in a plane. This is done simply by rotating the orientation vector at each time step so that it lies in the plane. Concretely, we calculate the orientation vector in the observer frame:

$$\widehat{\boldsymbol{p}} = q\widehat{(0, 0, 1)}\, \bar{q}$$

Then we rotate the orientation vector around the axis **u** that is orthogonal to both the $z$ axis and **p**: $\boldsymbol{u} = (0, 0, 1) \times \boldsymbol{p}$, by an angle $\theta = \sin^{-1}(\boldsymbol{p}_z/\|\boldsymbol{p}\|)$. The final orientation quaternion is then $q' = Q(\boldsymbol{u}, \theta)q$, where $Q(\boldsymbol{u}, \theta)$ is the rotation of axis **u** and angle $\theta$.

## Behavioral scenarios

In all simulations, the electrophysiological model fitted to the cell shown in Fig 7 is integrated with Euler method and a time step of 0.1 ms, using Brian 2 [45]. Kinematics were integrated with a time step of 1 ms for simulations with stimuli with sharp boundaries (Figs 7, 8, 8B and 9A) and 2 ms for simulations with spatially continuous stimuli (Fig 9C and 9D).

*Avoiding reaction*. In Fig 9D, models are simulated in a plane with 2 ms pulse currents of amplitude 0.3, 0.5 and 5 nA, at 1 second intervals. In Fig 7F, the stimulus is a 2 ms current pulse of 0.01 to 10 nA amplitude. In Fig 7G, the stimulus is a 100 pA pulse of duration 0 to 100 ms. The reorientation angle is calculated as the change in angle before and after the stimulus, averaged over 1 second (the spinning period).

*Interaction with a stimulus*. In Fig 10 and all subsequent figures, trajectories are constrained to a plane. In Fig 10A and 10B, the stimulus is a half-plane. It produces an instantaneous depolarizing current, proportional to the fraction of the cell shape inside the stimulus (see *Sensory transduction*), with maximum 5 nA. In Fig 10C and 10D, the stimulus additionally goes through a low-pass filter with time constant 40 ms, representing the activation/deactivation rate of the receptors.

*Repelling and attracting discs*. In Fig 11A and 11B, the stimulus is a disc of radius 1 mm within a 4 mm torus. 100 trajectories are simulated for 20 s, with random initial positions. A noisy current is added to the membrane equation so as to produce spontaneous action potentials at the observed rate of 0.2 Hz. It is modeled as an Ornstein-Uhlenbeck process:

$$\tau_n \frac{dI_n}{dt} = -I_n + \sigma_n \sqrt{\tau_n} \xi$$

with $\tau_n$ = 20 ms and $\sigma_n$ = 9 pA. Physiologically, this corresponds to the random opening of $K^+$ channels [109]. This noise is included in all subsequent simulations.

In Fig 11A, the stimulus produces a depolarizing current with a 40 ms time constant, as in Fig 8C and 8D. In Fig 9B, the stimulus produces an adapting hyperpolarizing current (see *Sensory transduction*; $\tau_{fast}$ = 40 ms and $\tau_{slow}$ = 200 ms), with a maximum amplitude of 1 nA.

*Gradient following*. In Fig 11C, the environment has the topology of a cylindrical surface, i.e., circular in the small dimension (500 μm) and linear in the long dimension (25 mm). The stimulus is a linear gradient of 100 pA/mm, with transduction modeled with adaptation as for the attracting disc (Fig 9B).

*Collective behavior*. In Fig 11D, cells produce $CO_2$ by respiration, which then diffuses and acidifies the fluid. This is modelled by the diffusion equation:

$$\frac{\partial S}{\partial t} = \alpha.1_{(x,y)\in\text{cell}} + D\Delta S$$

where $S$ is the transduction current triggered by $CO_2$, $\alpha$ is the production rate and $D$ is the diffusion coefficient of $CO_2$. In water at 25°C, $D \approx 0.002$ mm$^2$/s but we accelerate it by a factor 5 to speed up the simulation. The production rate is $\alpha$ = 100 pA/s in a square pixel of width 20 μm.

## Supporting information

**S1 Movie. Interaction of the Paramecium model with a depolarizing stimulus, corresponding to Fig 10A.**
(MP4)

**S2 Movie. Interaction of the Paramecium model with a depolarizing stimulus and slow kinetics, corresponding to Fig 10C.**
(MP4)

**S3 Movie. Model trajectories on a torus with a depolarizing circular stimulus, corresponding to Fig 11A.**
(MP4)

**S4 Movie. Model trajectories on a torus with an attracting circular stimulus, corresponding to Fig 11B.**
(MP4)

**S5 Movie. Model trajectories on a linear gradient (with circular topology in the vertical direction), corresponding to Fig 11C.**
(MP4)

**S6 Movie. Model trajectories on a torus with $CO_2$ production (breathing), diffusion, and chemosensitivity, corresponding to Fig 11D.**
(MP4)

**S7 Movie. Freely swimming paramecia.**
(MP4)

## Acknowledgments

We thank Marcel Stimberg for technical assistance and Eric Meyer for providing us with specimens of *P. tetraurelia*.

## Author Contributions

**Conceptualization:** Irene Elices, Romain Brette.

**Data curation:** Romain Brette.

**Formal analysis:** Romain Brette.

**Funding acquisition:** Léa-Laetitia Pontani, Alexis Michel Prevost, Romain Brette.

**Investigation:** Irene Elices, Anirudh Kulkarni, Nicolas Escoubet, Romain Brette.

**Methodology:** Irene Elices, Anirudh Kulkarni, Nicolas Escoubet, Léa-Laetitia Pontani, Alexis Michel Prevost, Romain Brette.

**Software:** Romain Brette.

**Supervision:** Léa-Laetitia Pontani, Alexis Michel Prevost, Romain Brette.

**Validation:** Irene Elices, Romain Brette.

**Visualization:** Irene Elices, Romain Brette.

**Writing – original draft:** Romain Brette.

**Writing – review & editing:** Irene Elices, Anirudh Kulkarni, Nicolas Escoubet, Léa-Laetitia Pontani, Alexis Michel Prevost.

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
