## [Decision Letter · Decision Letter 0]

25 Nov 2022

Dear Dr. Brette,

Thank you very much for submitting your manuscript "An electrophysiological and kinematic model of Paramecium, the “swimming neuron”" for consideration at PLOS Computational Biology. As with all papers reviewed by the journal, your manuscript was reviewed by members of the editorial board and by several independent reviewers. The reviewers appreciated the attention to an important topic. Based on the reviews, we are likely to accept this manuscript for publication, providing that you modify the manuscript according to the review recommendations.

Sincerely,

Joseph Ayers, PhD

Academic Editor

PLOS Computational Biology

Marieke van Vugt

Section Editor

PLOS Computational Biology

Reviewer's Responses to Questions

**Comments to the Authors:**

Reviewer #1: The review is uploaded as an attachment.

Reviewer #2: Summary

This work presents a complete biophysical characterisation of the action potential of the ciliate paramecium - once a key model for electrophysiology and behaviour and since forgotten. A major strength of the paper is its ability to discriminate between and measure (or infer) different kinetic parameters and ionic fluxes, such as through the use of a deciliation protocol. These carefully derived parameters are then used as input to a idealised swimming model to understand at a conceptual level the relationship between the membrane potential, ciliary orientation, asymmetry, and swimming trajectories. These integrative insights are supported in part by data, and extended to assess how the organism should respond to certain idealised environmental stimuli. The authors have also chosen to make public the codes and data associated with the paper which will make this a valuable contribution to the field.

This paper is very dense and contains a lot of information, this makes it a very hard read for anyone who is not familiar with the subject. It is difficult to keep track of the main message of the paper - the authors should consider simplifying some sections or removing some descriptions that do not necessarily add to the main narrative. The level of detail and rigour applied to the electrophysiological characterisation is not matched by the later sections on behaviour - this imbalance itself is due to many experimental limitations (which is fine to acknowledge) but can be addressed by structuring the paper in a more streamlined way.

For example the summary/discussions can be shortened, e.g. the section how paramecium turns appears to repeat some of the material from the results section. Descriptions of how specific model parameters lead to certain behaviours can be shortened/simplified - since there is probably over-interpretation of the model in some cases. In the case of the artificial scenarios considered, though these are interesting, the assumptions and equations behind them are not explained as carefully as the earlier sections of the paper.

1. Some structural issues:

It is also sometimes unclear which results/panels are from theory and which from experiments. Similarly, sometimes descriptions of results from previous work from other authors are merged with the current findings in the same paragraph (e.g. page 4 - ‘when calcium enters cilia,…’). If the journal format allows, the authors should consider restructuring parts of the paper so it’s more obvious which sections are results arising directly from their work and what is background info.

Another point is that since it was not possible to measure calcium concentration directly, a lot of references to ‘x happens when calcium increases/decreases’ are not supported by experimental data but are rather inferred, but the text sometimes makes this distinction ambiguous.

2. In general the figure captions are extremely long - can theses be shortened? Some of the information is repeated in the text.

3. The distinction between ‘escape reaction’ and ‘avoidance response’ is unclear, and the terms have been used interchangeably in other species - though the authors are careful to state that they are only dealing with the latter. Then how is the escape reaction different from the avoidance response in terms of behaviour and electrophysiological properties, what actually happens in the paramecium escape, this should be made clearer or include the relevant citations.

 4. In order to aid readability - perhaps there should be a figure panel summarising the idealised hydrodynamic model/whole-body approach? Perhaps one of the later figures should be reconfigured to make clear what is the role of the model and where it enters as part of the ‘integrative model’ - at the moment it’s hidden deep in the supplement.

5. In general the PIV data was much more noisy than the ephys signal, and also is this enough to justify the claim that the cilia beat direction reverses longer than the corresponding potential signal? Not sure if scales with the signal intensity either - look at the purple and red lines?

6. Given that kinetic parameters are matched so precisely in the early sections, why is it that ‘we simply set both the forward and backward maximum velocity at v = +/- 500 um/s?’ (Bottom of page 15). The behaviour sections seems to be treated less rigorously than the action potential modelling - is there good justification for this?

7. Some results according to Jennings or other older literature are referenced when it might have been possible to verify this from the new data, e.g. fig 6D - regarding the directional change in theta during the avoidance reaction, was this observed by the authors as well? What kind of stimulus method did Jennings use to get the different levels of avoidance response - and how does this relate to the injected current in the present study?

8. One consequence of this interpretation is that the paramecium is very sensitive to the position of the oral grove at the onset of the stimulus, does this make sense physiologically and was this observed in the experiments? Or could this be an artefact of the fact that the movement is constrained in the plane in the simulations? This makes trying to infer any relationship between backward swimming and reorientation angle quite difficult from 2D projections of the complete 3D movement.

9. Clarify difference between open and closed-loop behaviour Why is chemotaxis considered closed loop…? This will be helpful for non-experts.

10. Some later section on CO2 and formation of aggregations - this is not well-explained, more details should be given either in the supplement and make some clear reference to where to find this information in the text. Page 21 line 584 and also later in that section - ‘breathing’ is the wrong word here, maybe respiration? Breathing is normally reserved for organisms with lungs and a full respiratory system? Also in the adaptation section - two pathways with opposite polarity were assumed, but it’s unclear why? And also where the chosen time constants come from?

Other queries

Figure 1C - what’s the difference between the black and red arrows? Also in the caption - speaks of calcium current, but how do we know this?

Fig 4d - box plot data has a strange distribution gkd - what’s the reason for this?

Fig 5 - missing a colorbar showing which I (nA) values correspond to which lines.

Fig 6 - how does one define anterior and posterior from the videos? Is this done roughly by eye or is the orientation of the cell accounted for by some sort of segmentation algorithm?

In Panel E - the labels should be defined, or perhaps remove any labels to structures that are not relevant.

Fig 7- can the layout of the panels be rearranged to make it clearer which label refers to which panel, e.g. F, G are a bit confusing. And why does panel G not have the equivalent of the red line (different orientation of the oral groove) in panel F? Also where is the oral groove in the simulations/model - can this position be highlighted? How exactly is the reorientation angle defined in both sims and experiments? If in panel H the same quantities are plotted, then use the same variable names as in F,G? Colours not explained - presumably contour lines.

Reviewer #3: I enjoyed reading this paper and I think it is quite suitable for PLoS Comp bio. The authors have done a great deal of physiology and modeling to capture the biophysics of the paramecium action potential. They have made a hydrodynamic model in order to guide the creation of a kinematic description of the movement. They also built a clever closed loop version of the model that uses a simple shape to estimate the size of the stimulus (through the area stimulated). I have a few minor comments.

What or where is the oral groove?

If I understand it the kinematic model does not model the actual cilia movement but rather uses the calcium concentration in a space clamped model along with the three curve’s forms v,theta, omega to drive the paramecium

What accounts for the noise in the trajectories? Is this deterministic chaos, or is there noise added? Or is this just from the spinning? So it’s sort of an independent periodic behavior (spinning) decoupled from the linear motion except through the heuristic transfer functions in fig 7? In real cells is there any correlation between spinning and forward movement?

Regarding this, it would be interesting to see more details on how the turn is made depending on the phase of the spin (which i guess determines where the oral groove) If I understand it, the oral groove somehow braks the symmetry of the cell and the effects of stimuli are somehow related to this> I may be wrong in which case the authors need to do a better job of explaining how the oral groove alters the motion. So much of the details were buttied in the extremely long methods section.

On the equation for the calcium dynamics:Are the signs correct in this equation for calcium? I could not find a value for lambda, but it better be negative. I think a minus sign is needed for the calcium influx and if lambda is positive, then it is incorrect

**Have the authors made all data and (if applicable) computational code underlying the findings in their manuscript fully available?**

Reviewer #1: Yes

Reviewer #2: Yes

Reviewer #3: Yes

PLOS authors have the option to publish the peer review history of their article (what does this mean?). If published, this will include your full peer review and any attached files.

Reviewer #1: No

Reviewer #2: No

Reviewer #3: No

Figure Files:

Data Requirements:

Reproducibility:

References:

---

## [Decision Letter · Decision Letter 1]

26 Jan 2023

Dear Dr. Brette,

We are pleased to inform you that your manuscript 'An electrophysiological and kinematic model of Paramecium, the “swimming neuron”' has been provisionally accepted for publication in PLOS Computational Biology.

Best regards,

Joseph Ayers, PhD

Academic Editor

PLOS Computational Biology

Marieke van Vugt

Section Editor

PLOS Computational Biology

Reviewer's Responses to Questions

**Comments to the Authors:**

Reviewer #1: Summary:

The manuscript has improved, and the readability is much higher now compared to the initial submission. The authors have addressed all of my comments on the initial submission. In particular, the presentation of the electrophysiology fitting procedure is much more accessible and the current conventions, which I found confusing in the initial submission, are explained much better. I also think that the two additional figures (1 and 4) really help to guide through the manuscript. The authors should consider my three minor comments below, basically as a service to the reader to further improve clarity of the manuscript.

Minor comments:

1. Fig. 1C/D: Why does Paramecium stay inside the acid drop? Shouldn’t it try to avoid it? This is only explained later for Fig. 11, so it should be mentioned earlier.

2. Fig. 8D: I still find this figure confusing. In each subpanel, we see one animal turning, with non-zero omega and theta. Presumably, in a realistic setup, the animal would also move forward while it spins, so the upper two subpanels would change, so that the animal moves up (or down) while turning at the same time. Can it move forward if it rotates in the way shown in the lowest subpanel? I would think no. It would fit to the model results shown in 8F(3), where maximal theta means zero v and maximal omega. Please clarify this.

3. Please clarify how ‘…calcium might differentially activate the cilia…’ in the model. From Fig. 9, we see that v, theta and omega are all functions of the intracilliary calcium concentration. From the phenomenological hydrodynamic model, we know that cilia must activate differentially during a turn. How is this implemented in the model? I am not asking for how this is effected biologically (which is beyond the scope of the paper, as the authors say), but just how it is implemented in the model. I understand that the relations shown in Fig. 9 apply to the entire animal, and not to individual cilia, and a sweep in calcium concentration (from low to high) gives us the turning, as v changes sign and theta and omega peak when this happens. Yet, the sentence in the Discussion in ll. 742 ff. requires more explanation, in my eyes. Why is it ‘…the implicit assumption of our model.’? Because there is a critical calcium concentration for ciliary reversal that causes non-zero theta and omega, and that once we move beyond this concentration, theta and omega return to their pre-reversal values, whereas v changes sign?

Reviewer #2: I am satisfied that the authors have successfully addressed most of the comments of the reviews and that it is now ready for publication. (The new figure 4 is helpful, in the caption instead of 'each number indicates the corresponding figure' say 'each number indicates the corresponding figure below' just to be clear)

Reviewer #3: The authors have addressed my concerns

**Have the authors made all data and (if applicable) computational code underlying the findings in their manuscript fully available?**

Reviewer #1: Yes

Reviewer #2: Yes

Reviewer #3: Yes

PLOS authors have the option to publish the peer review history of their article (what does this mean?). If published, this will include your full peer review and any attached files.

Reviewer #1: No

Reviewer #2: No

Reviewer #3: No

---

## [Editor Report · Acceptance letter]

5 Feb 2023

PCOMPBIOL-D-22-01305R1 

An electrophysiological and kinematic model of Paramecium, the “swimming neuron”

Dear Dr Brette,

I am pleased to inform you that your manuscript has been formally accepted for publication in PLOS Computational Biology. Your manuscript is now with our production department and you will be notified of the publication date in due course.

With kind regards,

Dorothy Lannert
